

# The dependency of internal tides on background stratification variability : a case study on the Amazon shelf and the Bay of Biscay

Simon Barbot[1], Florent Lyard[1], Michel Tchilibou[1], and Loren Carrere[2]

[1]LEGOS, Université de Toulouse, CNES, CNRS, IRD, UPS, Toulouse, France
[2]CLS, Ramonville-Saint-Agne, France

**Correspondence:** Simon Barbot (simon.barbot@legos.obs-mip.fr)

**Abstract.** The forthcoming SWOT altimetric mission aims to access the smaller mesoscale oceanic circulation with an un-precedented spatial resolution and swath. The repetitivity of the mission orbit implies that high frequency processes, such as the internal tides (ITs), are under-sampled in time and their full temporal evolution is not observed. They are therefore aliased onto lower frequencies and possibly mixed into the mesoscale signals. As with the barotropic tides, the ITs must be
corrected from the altimetric observations in order to access to the smaller mesoscales. But unlike barotropic tides, ITs are not completely stationary and have significant temporal variability due to their interactions with the ocean circulation and the stratification variability. ITs prediction, correction and error calculation requires a precise understanding of the ITs' surface elevation signature and its temporal variability. Stratification changes impact both on the generation and the propagation of ITs. This present study proposes to quantify the impacts of the background stratification variations alone with a classification of the
observed stratification and an idealized modelling of the ITs. A single methodology is developed to handle a very broad range of stratification variability. The classification is made using clustering methods and the modelling uses the frequency domain model T-UGO.

The methodology is successfully tested on the Amazon shelf and in the Bay of Biscay. For the Amazon shelf, the pycnocline depth linearly impacts on the amplitudes and wavelengths of the ITs first two modes. An increase of the pycnocline depth
increases the total ITs' amplitude but also transfers the energy from the mode 2 to the mode 1. An increase of the pycnocline depth also increases the wavelengths of both modes 1 and 2. In the Bay of Biscay we found no such proxy to describe the changes in ITs' characteristics so a seasonal climatology is explored. The seasonality of the stratification strongly affects the amplitudes of modes 2 and 3 and significantly impacts on the surface elevation of ITs. Whereas the wavelengths of all modes and the amplitude of mode 1 are only weakly affected by the stratification seasonality. The amplitude variability of modes 2
and 3 also modifies the ratio between the modes in presence and makes the horizontal scales of ITs variable. The significance of the ITs wavelength modifications with stratification changes suggests that a more accurate ITs' surface elevation correction for SWOT measurements should take into account this stratification variability.



# 1 Concepts and objectives

## 1.1 Internal tides issues for altimetry

Internal tides (ITs) are internal waves with tidal frequency. ITs are generated when the energy fluxes of the barotropic tide are perpendicular to a bathymetric slope within a stratified ocean. The vertical oscillations produced interact with non-tidal ocean circulation and enhance the vertical mixing of the ocean. The generation and propagation of ITs are modulated by the ocean circulation, especially the mesoscale dynamics, but also by the stratification variability. Thus, contrary to barotropic tides, the sensitivity of the ITs to the ocean conditions is responsible for a significant temporal variability of their harmonic amplitude and phase.

The studies of Ray and Mitchum (1996, 1997), first around Hawaiian islands and then globally, were the first to observe ITs from altimetric time series using harmonic analysis. Thier method highlighted that the surface elevation due to ITs had a stationary signature over the 4 years of measurements. Ever since, the ITs' surface elevation has been separated into stationary (or coherent, or phase-locked in the literature) and non-stationary (or incoherent). The non-stationarity of the ITs is due to the temporal variability of the ocean conditions impacting the ITs' generation and propagation. The ITW' generation is impacted by the stratification changes due to radiative forcing or ocean circulation changes and by the variability of the barotropic tidal forcing. The ITs' propagation pathway is also impacted by such stratification modifications as well as circulation processes such as geostrophic currents (*e.i* Pereira et al., 2007 for realistic approach ; *e.i* Chuang and Wang, 1981 for idealized approach) and eddies (*e.i* Duda et al., 2018 for realistic approach ; *e.i* Ponte and Klein, 2015 for idealized approach) that directly interact with the ITs. With the increase in the altimetric time series, global maps of the stationary ITs' surface elevation amplitude become more precise (Carrere et al., 2004; Ray and Zaron, 2016; Zaron, 2019; Zhao et al., 2019; Carrere et al., 2021). Based on the residuals of these global maps, Zaron and Ray (2017) evaluated the non-stationary amplitude. The authors highlight that most of the tropics are dominated by non-stationary ITs.

The forthcoming SWOT satellite mission (Morrow et al., 2019) is designed to observe the fine scale 2D elevation of the continental waters as well as sea surface height (SSH). Global measurements will be made along two 50km wide swaths in addition to the traditional nadir measurements. During its nominal phase, SWOT's wide-swath coverage will be repeated every 20.86 days for at least 3 years (Fu and Ubelmann, 2014). This orbit was carefully selected to separate the main tidal constituents after 3-years. For the ocean, two products with a resolution of 2km and 250m should be available in the open and coastal oceans[1]. This wide-swath measurement pattern will allow us to access the fine scale SSH and, for the first time, observe the smaller mesoscale circulation and ITs in 2D (processes from 150km to 15km).

Such spatial resolution enables us to detect the ITs but the poorer temporal resolution prevents them from being properly resolved in frequency space. The high frequency signals of the ITs are aliased into lower frequencies, within the range of mesoscale and sub-mesoscale processes. For instance, the aliased period of the main three tidal constituents for SWOT will be about 66 days for the M2 tides, about 77 days for the S2 tides and about 266 days for the K1 tides (for T/P : 62 days for M2, 58 days for S2 and 270 days for K1). ITs' SSH wavelengths are also in a similar range as the typical spatial scales of

---

[1]https://podaac.jpl.nasa.gov/SWOT?tab=datasets§ions=about



mesoscale and sub-mesoscale circulations. It is important to separate the aliased barotropic tides and ITs signals from the SSH before calculating geostrophic currents or vorticity, since the tide signals are not in geostrophic balance. Yet the overlap in spatial and temporal variability between ITs and the mesoscales creates a complex separation issue. In harmonic analysis from altimetry observations, the contamination of tidal signal by non-tidal ones generally diminishes as the observation duration increases. For quasi-stationary tides (such as barotropic tides), this means that the accuracy of the tidal harmonics improves with time. However, for ITs, the proportion of the stationary component, captured by the harmonic analysis, in the total ITs' SSH diminishes as the observation duration increases due to the increased ocean variability over time. So empirical ITs corrections, are either inaccurate if based on short observation periods (stationary part issue) or incomplete if based on long observation period (non-stationary part issue).

The SWOT measurements cumulate two factors: an inappropriate space-time sampling to resolve ITs and the non-stationarity of the ITs themselves (Arbic et al., 2015). The interlaced space-time spectrum of ITs and mesoscale circulation in the SWOT observations advocate for producing the best possible ITs correction and then quantifying the ITs' residuals that will not be corrected. For these reasons an international effort is taking place in order to propose new methods of ITs detection in SWOT observations (*e.i.* Zhao et al., 2018) and increase the knowledge on ITs' non-stationarity (*e.i.* Tchilibou et al., 2019). The present study aims at contributing to the understanding of the ITs' non-stationarity.

One of the key factor of the ITs generation and propagation is investigated: the stratification and its temporal variability. A dual approach will be used based on the classification of observations and ITs modelling. The classification will help us to characterize the stratification variability. The ITs modelling will enable us to quantify the impacts of such stratification variability on the ITs' SSH. Such idealized simulations are interesting to explore the ITWs properties, before running more realistic but complex 3D simulations. This methodology will be tested in two areas where the stratification variability is driven by different processes. The study is divided in two parts: the stratification classifications (section 2) and the ITs modelling (section 3). More details about the ITs and the stratification can be found in the rest of this section.

## 1.2 ITs' dependency to stratification

Stratification is the restoring force of the internal waves. Starting from a stratified ocean at rest, with horizontal isopycnals, the vertical displacement of a given layer at some position will create a horizontal baroclinic pressure gradient. This instability will propagate as an internal gravity wave. Stronger stratification will generate stronger internal gravity waves. In the case of ITs, the vertical displacements occur when the periodic, barotropic (horizontal) tidal flow is oriented across a topographic slope. In a finite depth ocean, assuming that hydrostatic approximation holds, the ITs vertical wavenumbers are constrained by the depth. Those wavenumbers can be developed in different vertical modes for each permitted vertical wavenumber (the first mode being the barotropic one). In a continuously stratified ocean, the number of modes can be infinite. In a two-layer ocean, only one baroclinic vertical mode is supported. In a numerical model, the number of modes will be limited by the number of layers in the model grid.



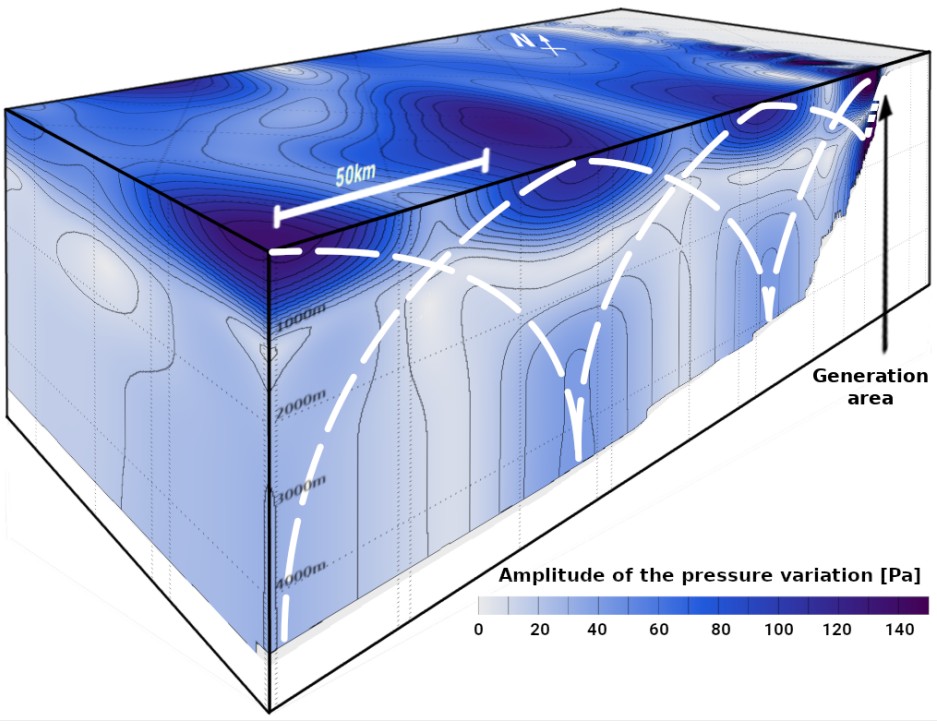

**Figure 1.** 3D visualisation of the internal tides (ITs) signature on the pressure amplitude. The white dashed lines represent the wave beam of the ITs.

The commonly used proxy for the stratification is the Brunt-Väissälä frequency $N$ (Gill, 1982):

$$N^2 = -\frac{g}{\rho_0}\frac{d\rho_0}{dz} \tag{1}$$

with $g$ the gravitational acceleration, $\rho_0$ the unperturbed potential density. $N$ is one of the governing terms of the linear internal wave theory that accounts for the velocity and pressure perturbations in the vertical equations. In practice, the vertical profile of N will dictate the partition of energy between the various modes (mostly concentrated in the first ones). For a given N profile, each mode has a specific horizontal wavenumber associated with its vertical wavenumber through the dispersion relationship (internal gravity wave in a rotating fluid Kundu et al., 2004):

$$\omega^2 - N^2\frac{k_H^2}{k_H^2 + k_V^2} + f^2\frac{k_V^2}{k_H^2 + k_V^2} = 0 \tag{2}$$

with $f$ the Coriolis parameter, $\omega$ the wave frequency, $k_H$ the horizontal wavenumber and $k_V$ the vertical wavenumber.

The variety of horizontal wavelengths results in a field of constructive and destructive interactions between the various modes composing the physical internal wave. The ITs' final signature is illustrated by Figure 1: where the amplitude is high, the modes interactions are constructive; where the amplitude is low, the modes interactions are destructive. The pathway of

high amplitude supports the energy flux of the ITs and is called the wave beam (white dashed lines on Figure 1).



Being waves, the ITs are sensitive to the density of the field they are propagating through. Spatial variability of the density causes the refraction and reflection of the waves. The ITs slope $a$ in the water column (equivalent to the slope of the wave beam) is define by the ratio between horizontal and vertical wavenumber, from the Equation 2 this leads to:

$$a^2 = \frac{k_H^2}{k_V^2} = \frac{\omega^2 - f^2}{N^2 - \omega^2} \tag{3}$$

Because $N$ depends on the depth, the vertical profile of $N$ implies a vertical profile of $a$ as well. The variation of $a$ with depth explain the wave beam shape, like a Gothic vault. The surface elevation of the ITs directly depends on the value of $a$ throughout the water column as well as the ocean depth. As shown in Figure 1, the distance between two maximums of surface elevation is related to the wave beam pathway. So the stratification and the depth both impact the horizontal wavelength of the ITs during its propagation.

The stratification is thus one of the controlling term of the intensity and horizontal patterns of the ITs' generation and propagation. The goal of the present study is to quantify the impacts of the stratification variations on ITs' characteristics. The impact on the generation is monitored by the amplitude. The impact on the propagation is monitored by the horizontal wavelength. To have a realistic range of stratifications, the stratification variability needs to be investigated first.

### 1.3   Stratification variability

Two main processes control the stratification variability in the ocean. The first one is the radiative forcing from the Sun that heats up the surface layers. The seasonality of this process depends on the latitude: in the tropic the seasonality is weak, in the high latitude the seasonality is strong. The second one is the circulation of the ocean and the induced mixing of the water masses. The variability of the circulation has multiple timescales: away from strong currents, the circulation at intermediate and deep depths is almost stationary, and the circulation at the surface is highly variable at seasonal to shorter timescales.

Because the ocean circulation affects the ITs propagation, the complexity of its impacts on the ITs is beyond the scope of this study. Even though the stratification will be derived from the circulation, the stratification will be investigated as stationary in order to prevent further interaction with the circulation. Such stratification is further named *background stratification*.

Two areas of interest have been chosen for their very different stratification variability and strong ITs generation. The Amazon shelf is well known to be dominated by non-stationary ITs signature (Magalhães et al., 2016; Zaron and Ray, 2017).

Located at the equator, the stratification is dominated by the circulation rather than the radiative forcing. The recent regional simulation of Ruault et al. (2020) will be used to validate the results. The Bay of Biscay is one of the most studied ITs generation areas. Located in the mid-latitudes and with weak ocean circulation, the stratification variability is dominated by the radiative forcing. The variability of the background stratification is investigated by making realistic classifications of stratification profiles from *in situ* measurements. Then, each class provides a typical stratification profile.

A common way to classify the background stratification is to make four seasonal means (based on the mean of three months ; further named three-month means) or monthly means. Such a method is easy to use but affects the realism of the profiles. The temporal mean erases the extremums of the profiles, does not consider spatial variability of the area and vertically smooths the profiles. Using different boxes within an area and using a small time interval can increase the realism of the profiles but leads





to processing of a huge number of mean profiles (further named typical profiles). To maintain the realism of the typical profiles
with only a few of them, the profiles are classified using clustering methods.

The clustering methods are based on the similarity of the profiles with each other to calculate an optimal classification. They
are used in many different disciplines (from sociology and biology to economics and astrophysics) and start to be applied on
the ocean's physical properties. In the last years, they have been applied to improve temperature and salinity climatologies
(Hjelmervik and Hjelmervik, 2013, 2014), to track water masses (Martin Traykovski and Sosik, 2003; Oliver et al., 2004), to
study the evolution of the oceanic desert areas (Irwin and Oliver, 2009) or identify the impact of eddies on the water masses
(Pegliasco et al., 2015). These methods can handle spatio-temporal variability and can highlight the potential patterns of the
extremum profiles.

The present study is divided into two parts. The section 2 addresses the usage of the clustering methods, the different methods
and the different configurations investigated. The classification of each area will be discussed and compared to the equivalent
three-month mean. Section 3 addresses the modeling of the ITs based on the typical profiles. An academic configuration of
the T-UGOm, a frequency domain model, is used for each typical stratification profile. The evolution of the surface elevation
amplitude and wavelength will be extracted from the simulations and some climatologies will be proposed. The results of the
Amazon shelf will be compared to the regional simulation of Ruault et al. (2020) and the altimetric ITs map of Zaron (2019).

## 2    The classification of the density profiles

### 2.1    Data

To study the variability of the density profiles, the CORA dataset is used (Coriolis Ocean Dataset for Reanalysis; Szekely
et al., 2016 provided by Copernicus monitoring service[2] and SEANOE; SEA scieNtific Open data Edition[3]). Different versions
are available but the latest versions (>5.0) are reprocessed data and provide only monthly-averaged data. In order to get a
larger number of profiles, the quality controlled data compiled in the version 4.3 is used. This version gathers all kinds of
measurements in the ocean sorted by date and instrument. Because density is targeted, only the instruments that measure
profiles of temperature and salinity at the same time are selected: Argo float, CTD, XCTD and moorings. The areas of interest
are defined as follows: for the Amazon shelf, from 5°S to 15°N and from 60°E to 35°E; for the Bay of Biscay, from 43°N to
48.5°N and from 10°E to 0°E (Fig. 2b,d). These individual profiles are used for the cluster analysis.

The typical density profiles derived from the clusters are compared to common three-month mean, averaged over the two
areas of interest and processed from the same dataset. In addition the clusters are compared to existing climatologies, also
averaged over the two areas. For the Bay of Biscay, BOBYCLIM is used (Bay Of BiscaY's CLIMatology ; Charraudeau,
2006 ; produced by the Ifremer [4]). This seasonal climatology uses the profiles in this area from 1862 to 2006 classified
into four seasons (three-month means), using a grid of $1/5^{th}$ of degree. For both the Amazon shelf and the Bay of Biscay,

---

[2]http://marine.copernicus.eu/

[3]https://www.seanoe.org/

[4]http://www.ifremer.fr/climatologie-gascogne/climatologie/index.php





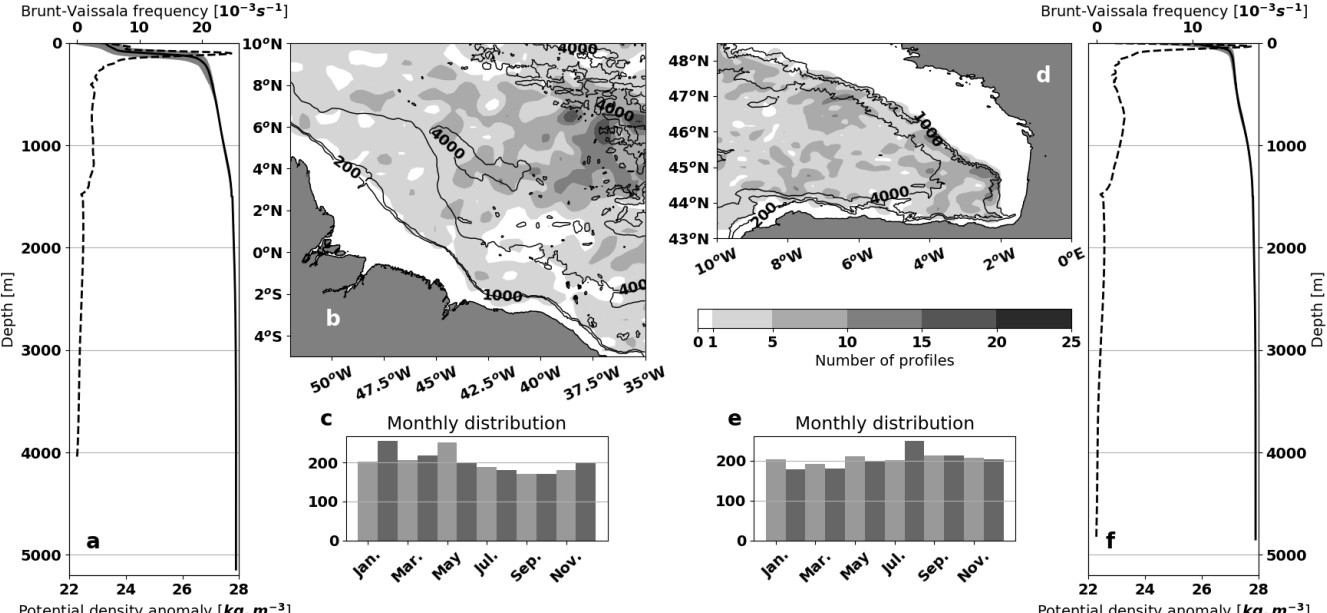

**Figure 2.** Characteristics of the *in situ* density profiles in the CORA V4.3 dataset for the two areas of interest: (a,b,c) the Amazon shelf and (d,e,f) the Bay of Biscay. (b,d) present the spatial distribution of the profiles. (c,e) present the monthly distribution of the profiles. (a,f) present the mean (solid line) and the 90% interval (grey patch) of the density profiles as well as the Brunt-Väissälä profiles (dash line).

ISAS13 climatology is used (*In-Situ* Analysis System ; Gaillard, 2015 ; provided by Copernicus and SEANOE). This monthly
climatology is based on the CORA database and averaged from 2004 to 2014, using a grid of $1/5^{th}$ of degree. The seasonal climatology of ISAS13 is built using three-month means. As the radiative forcing is weak in the Amazon shelf, the annual climatology is also build by averaging all the months.

Whatever the data, the potential density is calculated. The potential density is the density of the water that is adiabatically brought back to a reference pressure. This is equivalent to computing the density using the potential temperature instead of
the *in situ* one. This convention removes the effect of the pressure on the density. The ITs induce pressure oscillations so this consideration is absolutely crucial. Note that the potential density is responsible for the restoring forces and is used in the definition of $N$ (Eq. 1). The TEOS-10 convention (Millero et al., 2008) that uses the Gibbs Sea Water (GSW) equations of Feistel (2003, 2008) is used to calculate the potential density of the profiles (Python GSW package[5]).

Figure 2 presents the distribution of the dataset for both areas of interest. For the Amazon shelf area, the density profiles
(Fig. 2a) are strongly stratified around 100m. Most of the variability of the profiles is around this depth but also at the surface, mainly due to the variability of the North Brazil Current (NBC, Garraffo et al., 2003). The surface variability is also due to the Amazon river plume and the variability of the river discharge (Ffield, 2005). Then the stratification remains constant down to 1500m and slightly decreases close to the bottom. Note that the spatial distribution of the profiles is not homogeneous, less

---

[5]https://github.com/TEOS-10/python-gsw





profiles being available along the shelf. In the Bay of Biscay area, the density profiles (Fig. 2f) are less stratified than for the
Amazon shelf. The strongest stratification and the largest variability occurs at the surface. This pattern expresses the dominance
of the radiative forcing in this region. Around 700m, another stratification maximum can be observed. This particular pattern
is present in the study of Pichon and Correard (2006) and further detailed in Pairaud et al. (2010). The spatial distribution of
the profiles is quite homogeneous in this region.

## 2.2 Methodology of the classification

### 2.2.1 Pre-processing of the data

The variability of the ocean stratification is dominated by the surface density variations (Fig. 2a,d). Hence, in this section, the
profiles are selected and processed only for the surface layer, which is defined with respect to the variability of the profiles:
above 300m in the Bay of Biscay, above 600m in the Amazon shelf. Since the measurements have more uncertainties at the
surface, the top limit of the working layer is set to 10m.

The typical density profiles will be used in a frequency-domain tidal model (see section 3.1). Compared to time-stepping
models, this type of model does not support a stratification evolution over time and the simulated ocean needs to be at rest.
These concerns affects the density profile structure that the model can handle. If the density profile is unstable, high amplitude
instabilities are created because the stratification induces vertical motions rather than being a restoring force. Hence, only the
stable density profiles are processed. The measurements can still be noisy (presenting a negative density gradient) without
presenting unstable water masses. The threshold used to define an unstable profile is when there are more than two consecutive
occurrences of $\delta_z \rho < -0.5$ kg.m$^{-4}$ (with $\delta_z \rho$ the vertical density gradient).

    Focusing on the stratification sampling interval, the profiles need to have a decent vertical resolution. The selected profiles
must have more than 5 measurements over 100m inside the surface layer. In order to run the algorithms faster, the profiles need
to be on the same vertical grid and without any missing values. Thus, the density profiles are linearly interpolated to fill all the
gaps and get all the profiles on a vertical grid of 1m resolution.

    Now that the profiles are properly selected and interpolated, the shape of each profile needs to defined within a two co-
ordinates system in order to calculate the distance between each other. For that purpose, the profiles are processed using the
principal component analysis (PCA; Python SciKitLearn decomposition package[6]) following a similar procedure as Pauthenet
et al. (2018) in their temperature profile study within Southern Ocean fronts. Using this method, each profile is defined by its
projection on the two PCA axes. This new 2D plan containing all the profile projections is named PCA manifold. The clus-
tering method classifies the profiles within the PCA manifold by calculating the distance from one point to every other points,
minimizing the distance within each cluster. Once the classification is made within this artificial space, the classification can
be applied to the density profiles back on a physical space.

    The distribution within the PCA manifold can be improved by normalizing the profiles before performing the PCA. But as
expressed in Equation 3, the ITs wave beam slope $a$ is a function of $N$ that is directly affected by the value of the density $\rho_0$ in

---

[6]https://scikit-learn.org/stable/modules/decomposition.html



addition to the vertical gradient of $\rho_0$ (Eq. 1). So the ITs are directly affected by the values of the density profile, not only the shape of the profile. Hence, in order to keep the entire information of the profiles, the profiles are classified without using any normalization.

### 2.2.2 Parameters controlling the clustering methods

Three different methods of clustering have been tested: Ward, Average and Spectral (Python SciKitLearn clustering package[7]). Those methods have been selected because they can better classify this specific distribution of the PCA manifold. The Average hierarchical clustering method builds the complete tree that links the points by minimizing the average of the distances between the clusters being merged in order to build the tree (WPGMA, Sokal, 1958). The Ward hierarchical clustering method is based on the same methodology as Average but minimizes the variance of the clusters being merged (Ward, 1961; Ward and Hook,
1963). The Spectral clustering method is different from the previous two. This method projects the PCA manifold onto a polar coordinate space before performing the classification through a specified number of clusters and minimizing the distance within each cluster (Yu and Shi, 2003; Von Luxburg, 2007).

For these three methods, the sensitivity of two parameters needs to be investigated: the number of final clusters and the number of neighbors used in the calculation of the distance between profiles. For both areas, the methods are tested using
2 clusters to 10 clusters. The number of neighbors is important to properly manage the profiles that are isolated outside the PCA manifold. If the number of neighbors is weaker than the number of outsider profiles, then they all would be grouped in a dedicated cluster. Otherwise, they would be included in the cluster of the nearest profiles. This latter case can lead to groups of profiles that do not have the same shape inside the same cluster. The number of neighbors also affects the profiles located at the boundary between two clusters: depending on the number of neighbors, they would be included in one cluster or another. This
sensitivity is not the same for all the methods, and particularly the sensitivity is very low for the Ward and Average methods but bigger for the Spectral method.

After a wide range of tests, the Spectral and the Ward methods produce the best results. For the stability of the results, the Ward method is used in the rest of the study. The classifications using the 16 nearest neighbors are distributed more equally between the clusters so this parameter is chosen. The number of clusters is set more arbitrarily. For the Amazon shelf, the
variability of the density profiles is so simple that even 2 clusters could be enough to characterize the variability of the density profiles. For the Bay of Biscay, 10 clusters are not enough and leads to have some clusters with very few profiles. For both areas, a classification of 6 clusters is a good compromise that enables us to detail the evolution of the density profiles keeping well represented clusters.

### 2.2.3 Post-processing of the clusters for modeling purpose

Once the classification is done, the typical profile from each cluster is calculated in order to use it as a forcing in the simulations (see section 3.1). As explained above, the density profiles need to be stable (strictly increasing with depth) in order to get a

---

[7]https://scikit-learn.org/stable/modules/clustering.html





**Table 1.** Composition of the clusters and the characteristics of their stratification maximum for the Amazon shelf and the Bay of Biscay. The depth of $N_{max}$ corresponds to the pycnocline depth.

| | Amazon shelf (AS) | | | Bay of Biscay (BB) | | |
|---|---|---|---|---|---|---|
| Cluster | Profiles | $N_{max}$ | Depth $N_{max}$ | Profiles | $N_{max}$ | Depth $N_{max}$ |
| # | # | $[[10^{-3}\mathrm{s}^{-1}]]$ | [m] | # | $[10^{-3}\mathrm{s}^{-1}]$ | [m] |
| 1 | 628 | 25.7 | 72 | 787 | 4.0 | 133 |
| 2 | 576 | 22.6 | 108 | 424 | 11.3 | 35 |
| 3 | 528 | 23.7 | 88 | 419 | 7.2 | 45 |
| 4 | 336 | 23.9 | 128 | 324 | 19.9 | 35 |
| 5 | 336 | 23.8 | 148 | 316 | 19.3 | 35 |
| 6 | 17 | NA | NA | 177 | 15.9 | 30 |
| Mean | - | 23.9 | 106 | - | 15.2 | 52 |
| Total | 2421 | - | - | 2447 | - | - |

proper simulation. The density profiles also need to be defined at every depth of the configuration whose maximum depth is 4000m.

First, the median is calculated for each cluster. The clusters that do not have deep measurements are completed with the
median of all the profiles, no matter the cluster. If any of the profiles is deep enough, then the density is extrapolated with respect to the density gradient of the latest 4 measurements. The density gradient used for the extrapolation needs to be weaker than $5.0 \cdot 10^{-7}$ kg.m$^{-4}$ that is a common gradient at such a deep depth. The modelling of ITs does not support strong vertical gradient of density, so the median profiles have been smoothed using several forward-backward filters (Python SciPy signal package[8]). Many trials have been made to have the smoothest profiles while keeping most of the vertical patterns. From the
surface to 400m, the filter is a Gustafsson filter (Gustafsson, 1996) of order 3, with a critical wavelength of 25m. From 400m to 1500m, the filter is a zero-pad filter of order 2, with a critical wavelength of 100m. From 1500m to the bottom, the filter is a zero-pad filter or order 2, with a critical wavelength of 1000m. To be sure that the profiles are strictly stable, they are linearly interpolated on a high resolution grid of 0.5m and then sorted.

## 2.3 Application on the two areas of interest

### 2.3.1 Stratification of the Amazon shelf

The clustering method is performed on the Amazon shelf profiles and the 6 clusters (named AS-#) are sorted by the number of profiles they gather. The density profiles are measured from 1984 to 2015. The distribution of the 2421 profiles into the 6 clusters is detailed in the Table 1. Note that AS-4 and AS-5, the two deeper pycnocline clusters, have fewer profiles than the other clusters.

---

[8]https://docs.scipy.org/doc/scipy/reference/signal.html



**Figure 3.** Classification of density profile in the Amazon shelf in 6 clusters: (a) the PCA manifold of the profiles, (b) the cumulative proportion of the clusters during a mean year, (c) the spatial distribution (d) the median and the 90% interval of each cluster, (e) the measurement dates of the density profiles (the angle represent the day of the year and the distance from the center, the year of measurement). The colors of the clusters are common to all the graphs. The colored contours of (c) are set to highlight the areas gathering from 2 to 5 profiles (light color) and over 5 profiles (bold color) for each cluster. The black contours of (c) show the 200m, 1000m and 4000m isobaths.





Figure 3d illustrates the median of the different typical profiles obtained for each cluster. AS-6 contains only the density profiles that are exceptional: those 17 profiles show an offset of $1\ \mathrm{kg.m}^{-3}$ over the entire depth, and were measured at the same period (from October 2005 to March 2006), equally spaced by almost 10 days (Fig. 3e). These measurements have been made by a single ARGO float, (WMO number: 41953). This ARGO float failed its salinity measurements from the cycle 150 to 152 and from the cycle 166 to 180 (except the cycle 176) which explain the bad calculation of the associated density.

The clustering methods are efficient enough to detect those exceptional profiles that pass the standard quality control and can be used as a tool to filter them out. Gathering the suspicious profiles in AS-6 helps to sort the data and analyse only the realistic profiles contained in the other clusters.

Figure 3d illustrates that the variability of the Amazon shelf profiles is dominated by the depth of the pycnocline. This variability corresponds to the large 90% interval observed at the pycnocline depth in Figure 2a so this classification does

capture the realistic variability of the density profiles. The median value of the upper surface density is centered around 1023.3 $\mathrm{kg.m}^{-3}$ for all the clusters. Only AS-1 and AS-3 show a greater variability at the surface with the 90% interval (up to 1021.3 $\mathrm{kg.m}^{-3}$ and 1023.8 $\mathrm{kg.m}^{-3}$). All the typical profiles have a maximum stratification within the same range (Tab. 1), only the pycnocline depth differs. So, these clusters represent a good framework to investigate the influence of pycnocline depth on the ITs. From now on, the pycnocline depth will be used as a proxy to evaluate the influence of the ITs in this area.

The temporal variability of the clusters (Fig. 3b,e) shows that every cluster happen all the year. There is a seasonality very noisy due to the complexity of the circulation, its spatial distribution and its seasonality (explained below). The cluster classification enable to focus on a simple parameter (the pycnocline depth) rather than being blurred by the noise of a classical seasonal average classification.

AS-1 is the cluster with the shallower pycnocline depth. The stronger surface stratification of the 90% interval could be due

to the Amazon river plume. Indeed, the spatial distribution of AS-1 (Fig. 3c) corresponds to the area of the Amazon river plume during the retroflection events, usually from July to December (Ffield 2005, Figure 10). This does explain the occurrence of AS-1 from September to January but does not explain its occurrence from January to August (Fig. 3b). During this latter period the North Equatorial Counter Current (NECC, eastward) is weaker and the North Equatorial Current (NEC, westward) is a bit stronger (Richardson and Reverdin, 1987). In addition of the water masses from the Amazon plum, AS-1 could account

for the water masses of the NEC with a pycnocline around 70m but without the surface stratification. The distribution of the seasonality of AS-1 (Fig. A1) confirms the duality of the seasonality within this cluster: the water masses from the retroplection of the Amazon plum occur during September to February north of 5°N and the NEC water masses occur during May to July south of 5°N.

AS-2 and AS-3 are similar: they have the same seasonality , the same spatial coverage and gather the profiles with a pycno-

cline depth from 80m to 110m. The North Brazil Current (NBC) is highly influenced by large anticyclonic eddies that deepen the pycnocline in their center. These eddies are wind-driven from August to November and enhance the retroflection of NBC water masses into the North Equatorial Counter Current (NECC, Johns et al. 1998). AS-2 and AS-3 occur mainly from January from September, exactly when the NBC does not present eddies. The spatial coverage of these clusters is quite large and almost affect the entire area of interest. So these clusters identify the profiles distinctive of the steady state of the NBC.





AS-4 and AS-5 occur from August to November, so they have the same seasonality of the NBC eddies. These clusters gather the profiles with the deeper pycnocline. Thus they identify the profiles corresponding to the deepening of the pycnocline due to the large anticyclonic eddies of the NBC.

Garraffo et al. (2003) studied the same area looking at the different transport of water using a regional model. They separated the area in four sub-domains (Garraffo et al., 2003, Figure 11c), where two of them correspond to the area considered in the
present study. They highlight that the sub-domain of AS-2 to AS-5 (Garraffo et al., 2003, green in Figure 11c) is influenced by the southern waters coming from the NBC. The sub-domain of AS-1 (Garraffo et al., 2003, pink in Figure 11c) is more influenced by the NECC waters than the NBC waters. The cross-shelf transect from mooring measurements (2004, Figure 2 around 47°W) clearly shows the separation of NBC water masses, along the shelf, and the North Equatorial Counter Current (NECC) water masses off the shelf. The isopycnals' depth shows a difference of 100m between the two waters. This difference
is comparable to the difference of the pycnocline depth observed between AS-1 around 70m and AS-4 and AS-5 around 140m. Goni and Johns (2003) used a two-layers model to convert altimetric SSH to the upper layer thickness. The authors show that the anticyclonic eddies in the NBC could increase the upper layer thickness from 20m to 40m (Goni and Johns, 2003, Figure 10). This difference is comparable to the difference of the pycnocline depth observed between AS-2 and AS-3 around 100m and AS-4 and AS-5 around 140m.

### 2.3.2  Stratification of the Bay of Biscay

The clustering method is also performed on the Bay of Biscay profiles. The 6 clusters are sorted by their number of profiles. The density profiles are measured from 1991 to 2015 and the separation of the 2447 profiles into the 6 clusters can be found in the Table 1. In this dataset, no suspicious profiles are detected with the clustering method, so all of the clusters will be used for the density study. Note that BB-1 has the most profiles, BB-6 has the least and the other clusters are almost equally represented.
Figure 4d shows the 6 typical profiles processed from the 6 clusters. In this area, the main variability of the profiles is dominated by the upper surface density. BB-2 and BB-6 are the only clusters to have almost the same surface density. The difference between these two clusters is the pycnocline's depth.

Figures 4b and 4e highlights that the classification corresponds to the seasonality of the profiles. Chronologically, BB-1 corresponds to winter and spring conditions with deep mixed layers with quasi-homogeneous density profiles. BB-6 corresponds
to early summer with density profiles that linearly decrease up to the surface. BB-4 and BB-5 correspond to late summer and early autumn conditions with the most stratified profiles. Finally BB-2 closes the loop corresponding to late autumn with deep surface layer profiles. BB-4 and BB-5 cover the same period simultaneously and the differences are due to the intensity of the stratification: BB-4 corresponds to mild summer stratification generally in July-August and BB-5 corresponds to stronger summer stratification with more profiles in September-October.
There is also a transitional group, BB-3, composed of profiles from both before and after BB-1 (winter): in December and in May. BB-3 is designated as the shoulder season in the rest of the study. These profiles also occur during late winter corresponding to some heating events that start to build a stratification without establishing it.


**Figure 4.** Classification of density profile in the Bay of Biscay shelf in 6 clusters: (a) the PCA manifold of the profiles, (b) the cumulative proportion of the clusters during a mean year, (c) the spatial distribution (d) the median and the 90% interval of each cluster, (e) the measurement dates of the density profiles (the angle represent the day of the year and the distance from the center, the year of measurement). The colors of the clusters are common to all the graphs. The colored contours of (c) are set to highlight the areas gathering from 2 to 5 profiles (light color) and over 5 profiles (bold color) for each cluster. The black contours of (c) show the 200m, 1000m and 4000m isobaths.





The spatial distribution shows the cluster are almost equally distributed in the area (Fig 4c. BB-5 occurs more in the southeast of the area but it is also present elsewhere. This result confirms that the variability of the density profiles is dominated by radiative forcing rather than complex changing circulation pattern in this region.

As expected, the clustering methods do identify the seasonality contained in the mid-latitude variability. This classification separates the seasonal changes more distinctly than a simple three-month means: BB-1 lasts for 4 months, BB-6 lasts for 1 month, BB-4,5 last for 3 months simultaneously and BB-2 lasts for 2 months. Further comparisons are shown in the next section.

## 2.4 Discussions

In this section, some typical profiles from the clusters will be compared to seasonal climatologies (ISAS13 and BOBYCLIM) and to the three-month means made from CORA V4.3 dataset. For the Amazon shelf, the two extreme clusters are chosen: AS-1 and AS-5. AS-1 is compared to the spring mean because this is the season with the shallower pycnocline and because this cluster is highly represented in spring. AS-5 is compared to the fall mean because this is the season where this cluster occurs the most. For the Bay of Biscay, BB-2, BB-4 and BB-5 are used to investigate the influence of fewer months in the classification and to discriminate between mild and stronger events. BB-2 is compared to the fall mean and BB-4 and BB-5 are compared to the summer mean.

The climatology profiles are averaged on the same areas and smoothed following the method explained in section 2.2.3.

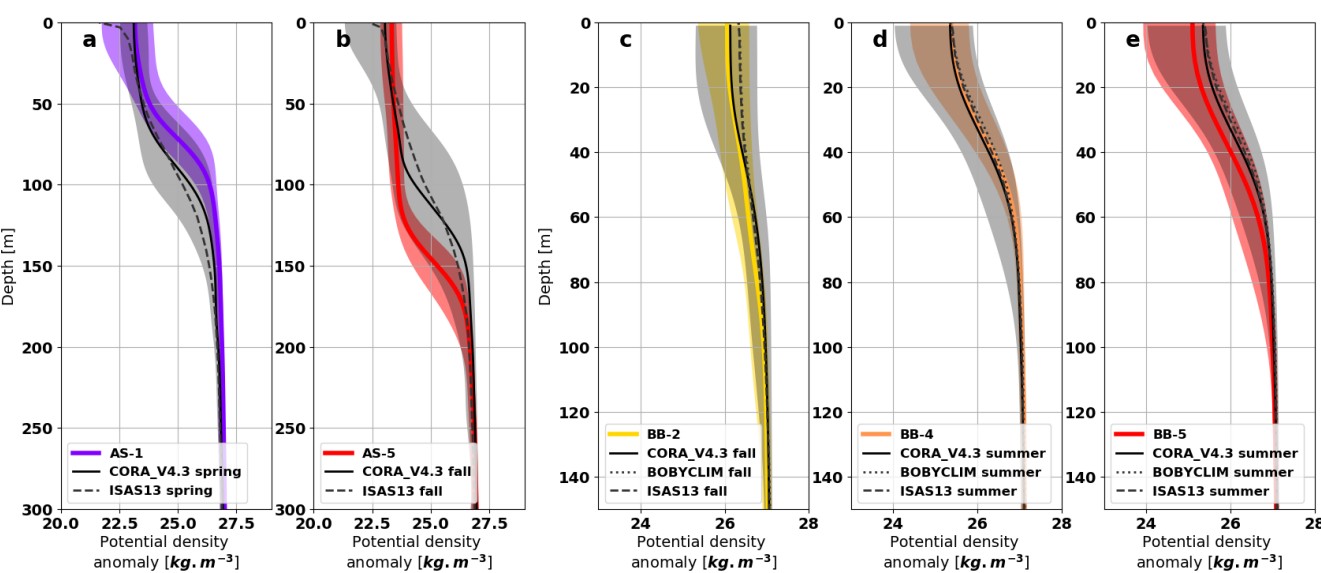

**Figure 5.** Climatology profiles compared to the corresponding selected typical profiles from the cluster classification: the Amazon shelf (a) AS-1 and (b) AS-3, the Bay of Biscay (c) BB-1 (d) BB-4 and (e) BB-5. The grey shadings are the 90% interval calculated from CORA V4.3 dataset using seasonal and the colored shadings are the 90% interval of each cluster.





For the Amazon shelf area, Figures 5a and 5b show that ISAS13 seasonal profiles are smoother than the CORA V4.3
seasonal profiles. After verification, the filters applied on the profiles cannot explain such differences. This difference can
likely be explained by the different periods used to produce these climatologies (2004-2014 for ISAS13, 1950-2015 for CORA
V4.3) and the gridding processes used for the ISAS13 climatology.

As expected from the climatology, the fall profiles shows a deeper pycnocline than the spring one (Fig. 5a and 5b). The
climatology profiles shows a slightly smoother pycnocline than the cluster profiles. Thus, the averaging of the large diversity
of the profiles within one season tends to smooth the stratification and does not represent it as well as the cluster classification.
AS-1 and AS-5 both represent a more contrasted part of the 90% interval of the seasonal profiles considered (grey patch).
The shallow pycnocline cluster AS-1 (Fig. 5a) sets the upper limit of the 90% interval of the spring climatology. The deeper
pycnocline cluster AS-5 (Fig. 5b) sets the lower limit of the 90% interval of the fall climatology. Still, the median of both
clusters is part of the seasonal 90% interval, indicating that the clusters represent a right proportion of these seasonal profiles.

These three-month means are not highlighting the profile variability as well as the cluster classification can do. As shown
above (Fig. 3c), the Amazon shelf has a strong spatial variability that explains why averaging classifications are ineffective.
The three-month mean is not recommended for this area that has complex circulation and mixing processes as well as a weak
dependency to radiative forcing.

For the Bay of Biscay area, the BOBYCLIM and ISAS13 climatologies are almost identical for the fall and summer seasons.
For fall, ISAS13 profile have a small difference of 0.2 kg.m$^{-3}$ at the surface compared to the climatology profiles made from
CORA V4.3. Otherwise the CORA V4.3 seasonal mean seems representative of the Bay of Biscay climatology.

The fall climatology profiles of CORA V4.3 and BB-2 profiles are very close (Fig. 5c), the overlap of both 90% intervals is
high, validating that BB-2 corresponds to the fall climatology.

The cluster classification highlighted two different clusters that correspond to the summer season: BB-4 corresponds to
mild summer stratification and BB-5 correspond to stronger summer stratification. Both clusters are compared to the same
summer climatology. The BB-4 median profile fit exactly the summer climatology profile. The 90% interval of the BB-4 shows
more stratified profiles than the summer climatology ones, especially shallower than 40m. The BB-5 profiles are indeed more
stratified than the mean summer climatology profile but better fit the 90% interval of the climatology deeper than 40m that are
not represented by BB-4. The main difference between the BB-4 and BB-5 is not located at the surface but around 60m where
the two 90% intervals of the clusters do not overlap each other. BB-4 and BB-5 contribute to different parts of the summer
climatology 90% interval and enable us to separate two stratification patterns that are mixed within the classical three-month
mean.

As expected, the cluster classification is more effective than the three-month mean to distinguish the different stratification
regimes that can occur within a given time period. The cluster analysis enables us to describe different pycnocline states:
established or transitory states and mild or extreme states. In the mid-latitude Bay of Biscay, the seasonal radiative forcing
is strong and makes the stratification uniform horizontally There, the cluster classification is more concise than the monthly
one: especially for the winter period, the 4 months between January to May can be considered as one group. In the Amazon
tropical regions, the spatial variability is more important due to a stronger interaction from the ocean circulation. This adds



**Table 2.** Inputs of COMODO-revised simulations

| Name | Variable | Value |
|---|---|---|
| Horizontal diffusion | $k_H$ | $1.10^{-3} \mathrm{m}^2.\mathrm{s}^{-1}$ |
| Vertical diffusion | $k_z$ | $1.10^{-3} \mathrm{m}^2.\mathrm{s}^{-1}$ |
| Roughness length | $z_0$ | $3.10^{-3} \mathrm{m}$ |
| Barotropic tidal velocity | $u$ | $10 \ \mathrm{cm.s}^{-1}$ |
| | $v$ | $0 \ \mathrm{cm.s}^{-1}$ |
| Relaxation length | $L$ | 42.5km |
| Relaxation time scale | $\tau$ | 72min |

complexity to the study of the stratification variability via a time dependent classification, and requires a good knowledge of the

region's circulation and water masses. As the cluster analysis does not preferably consider time dependent or space dependent classification, this is a better tool to investigate the complex variability within the tropics.

The clustering classification is used over a long period of time in this study, but doing so does not blur the inter-annual variability. The long term variability can be observe looking at the variability within the distribution between the clusters. Figure 3e and 4e can also help to observe such variability. For example, AS-4 and AS-5 are usually associated with the period

from August to November, but for the year 2006, AS-4 and AS-5 are only present from August to September.

## 3 Sensitivity of the internal tides to the background stratification

### 3.1 Model configuration for the ITs simulations

The T-UGOm (Toulouse Unstructured Grig Ocean model) has been used to simulate the ITs. Initially developed to resolve the two dimensional tidal equations (Piton et al., 2020), this model has been updated to resolve the three dimensional tidal equations

in the frequency domain (Nugroho, 2017). The model configuration is set to be hydrostatic and with a free surface. The 3D version uses Lagrangian layers that follow the fluid displacement in the vertical dimension. The experiments are focused on the M2 major tidal component in the two areas of interest and are based on the stratifications described in the previous section. The frequency domain calculation uses the tidal dynamical equations expressed in the complex, frequency space. This allows for much faster computation time than the time-stepping calculation. Studying interactions with non-tidal dynamics can be more

difficult, however our study aims to understand the influence of the background stratification alone on the ITs generation and propagation for each classified states. Thus, the frequency domain calculation option remains fully relevant.

All of the simulations are carried out with the same configuration and the same inputs that are shown in the Table 2. The reference latitude $\theta_{\mathrm{ref}}$ for the calculation of the Coriolis parameter is set differently for each area: $\theta_{\mathrm{ref}} = 0°\mathrm{N}$ for the Amazon shelf and $\theta_{\mathrm{ref}} = 47°\mathrm{N}$ for the Bay of Biscay. This enables us to compare the simulations with realistic cases. A single

density profile is used to set the stratification uniformly over all of the domain. The typical density profiles from the above




classifications (sections 2.3.1 and 2.3.2) are used and one simulation is made for each profile. As explain before, the density profile needs to be strictly stable because the frequency domain calculation does not allow the density to adjust like for the time stepping calculation.

This model is applied using the academic configuration from COMODO project (Ocean Modelling Community, 2011-2016, PI: L. Debreu, Soufflet et al., 2016) for the study of the internal waves generated on a continental slope (Nugroho, 2017). The project was originally built to compare different ocean models and T-UGOm 3D was one of them. The original configuration is based on the configuration of Pichon and Maze (1990): a flat bottom ocean of 4000m depth in the abyss (on the left) and 200m depth on the shelf (on the right); the domain is wide of 880km along the axis $x$ and large of one horizontal mesh, equal to 1km. The slope is described by the equations:

$$
\begin{cases}
\frac{d^2b}{dx^2} = -0.5\left(1 - \frac{cos(\pi(x-x_0))}{x_1-x_0}\right) & \text{for } x_0 < x < x_1 \\
\frac{d^2b}{dx^2} = -1 + 0.5\left(1 + \frac{x_2-x_1}{x_3-x_1}\right)\left(1 + \frac{cos(\pi(x-x_2))}{x_2-x_1}\right) & \text{for } x_1 < x < x_2 \\
\frac{d^2b}{dx^2} = 0.5\frac{x_2-x_0}{x_3-x_1}\left(1 + \frac{cos(\pi(x-x_2))}{x_3-x_2}\right) & \text{for } x_1 < x < x_2
\end{cases}
\tag{4}
$$

where $b$ is the bathymetry, $x_0 = 426$km, $x_1 = 443$km, $x_2 = 479$km and $x_3 = 484$km. The domain is described by 1760 Finite Element triangles using LGP1xLGP0 convention. LGP1 refers to the summit of the triangles where the pressure and elevation are set continuously from one triangle to another (Lagrange Finite Element of 1 degree of freedom). LGP0 refers to the barycenter of the triangle where the velocity is set (Lagrange Finite Element of 0 degree of freedom). On the vertical dimension, density is piecewise linear (*i.e.* linear inside layers with possible discontinuities at layers' interfaces), and velocity is uniform. The model is based on the primitive momentum equations, continuity equation and density advection equation. The model unknowns are the level displacements (including the free surface), horizontal velocities and density anomalies (due to advection in layers). However, a 3D wave-equation approach allows us to form a linear system where unknowns are limited to level displacements and density anomalies, velocities being then deduced once the 3D wave-equation system is solved.

This configuration places the slope in the center of the domain but this is not the optimal configuration for the present ITs study. As explained in the introduction, the wave beam slope is controlled by the stratification until it reaches the bottom. So the depth also controls the wavelength of ITW. This configuration only allows us to resolve around 2 or 3 times the wavelength in the abyss domain whereas more than 20 or 30 times the wavelength is resolved in the shelf domain. To compensate for this difference, the slope is shifted toward the shelf by 220km. This allows us to resolve around 4 or 5 times the wavelength in both domains.

After taking care of the horizontal resolution of the ITs, the vertical resolution needs to be investigated too. As shown in the previous section, the density profiles are steeper at the surface than at the bottom of the ocean. This means that the dynamics of the ITs are more different from one layer to another near the surface than at the bottom. Hence, a finer resolution at the surface is needed in order to resolve those dynamics. The vertical distribution of the 80 $\sigma$-layers (which follows the bathymetry) is set following a surface cosine: the layer thickness is thinner at the surface and larger at the bottom, the decrease is set by a cosine between 0 and $\pi/2$. Using $\sigma$-layers helps to get a strong resolution in the generation area, at the top of the shelf.





The Karman-Prandtl equation is used to calculate the bottom velocity affected by the bottom friction, as described for the AMANDES tidal model in the Amazon estuary (Le Bars et al., 2010). Using a frequency domain calculation, there is no time-step to set the bottom friction. So, an iterative process is used in order to make the bottom velocity converge, solving the

equations 4 times, each time using the previous bottom velocity. The model uses logarithmic buffer areas at the open boundaries (both sides) in order to stabilize the results. The relaxation term $R$ is expressed as follow:

$$R = \frac{exp\left(-\frac{d}{L}\right)}{\tau} \tag{5}$$

with $\tau = 72\text{min}$ being the relaxation time scale, $d$ the distance from the boundary and $L$ the relaxation length. In order to prevent the energy being reflected at the boundary, the relaxation length is extended from 20.0km to 42.5km on both sides.

In order to separate properly the baroclinic tides (ITs) from the barotropic tides, the solution is decomposed into vertical modes following the methods described in Nugroho (2017). For the following discussion, mode 1 and higher will refer to vertical baroclinic modes whereas mode 0 will refer to the vertical barotropic mode.

### 3.2 Modeling results on the two areas of interest

#### 3.2.1 Impacts of the Amazon shelf stratifications

As explained in section 2.3.1, the typical profiles from the Amazon clusters have almost the same stratification maximum and only the pycnocline depth differs in this area. From now on, these clusters profiles will be sorted by the depth of the pycnocline: AS-72m, 88m, 108m, 128m, 148m (corresponding to AS-1, 3, 2, 4, 5; Tab. 1).

Before detailing the impacts of pycnocline depth on the surface elevation, its influence on the baroclinic vertical modes is investigated first. Figure 6 (left panels) illustrates the three first baroclinic modes for all of the typical density profiles. The

depth of the extremum of all modes are influenced by the pycnocline depth (the black dots on the plots).

First, concerning the modes of the vertical structures $w$ and $\eta$. For mode 1, the deeper the pycnocline, the more the mode is shifted toward the surface. For mode 2, the deeper the pycnocline, the more the mode is shifted toward intermediate layers: the first extremum is deeper and the second one is shallower. For mode 3, the pycnocline depth only affects the first extremum: with a deeper pycnocline, the extremum is deeper. The same observation can be made for the modes of horizontal structures $u$,

$v$ and $P$. At the surface, with a deeper the pycnocline, the mode 1 is stronger and the higher modes are weaker. The impact of the pycnocline depth on the mode shifts seems linear for mode 1 and 2.

The only exception to this trend concerns mode 3 at the surface for the horizontal structures: AS-70m and AS-85m have the same amplitude, where AS-85m was expected weaker. This difference can be due to small density differences at the surface between the typical density profiles of about $0.02\text{kg.m}^{-3}$ which is the smallest difference at the surface between the clusters.

Figure 6 (right panels) illustrates the simulated amplitude of the baroclinic surface elevation. For the Amazon shelf simulations, the overall amplitude of the baroclinic elevation scales from 7cm to 10cm with no contribution of the modes higher than 2. When the pycnocline deepens, the total amplitude of the elevation increases with a domination of mode 1 over the mode 2.

**Figure 6.** Baroclinic modal structures for the three first modes (left panels) and simulated amplitude of baroclinic surface elevation for the five first modes (right panel) for all the typical density profiles of the Amazon shelf. The modal structures are different for vertical processes ($w$ and $\eta$, upper left panels) and horizontal ones ($u$, $v$ and $P$, lower left panels). The black points show the extremums of the modes. The simulations are sorted with respect to the depth of the pycnocline. On the right panels, the white line represents the sum of the baroclinic modes, the colored patches represent the part of each mode in the sum: the modes on top of the sum line refer to destructive interaction between the modes.





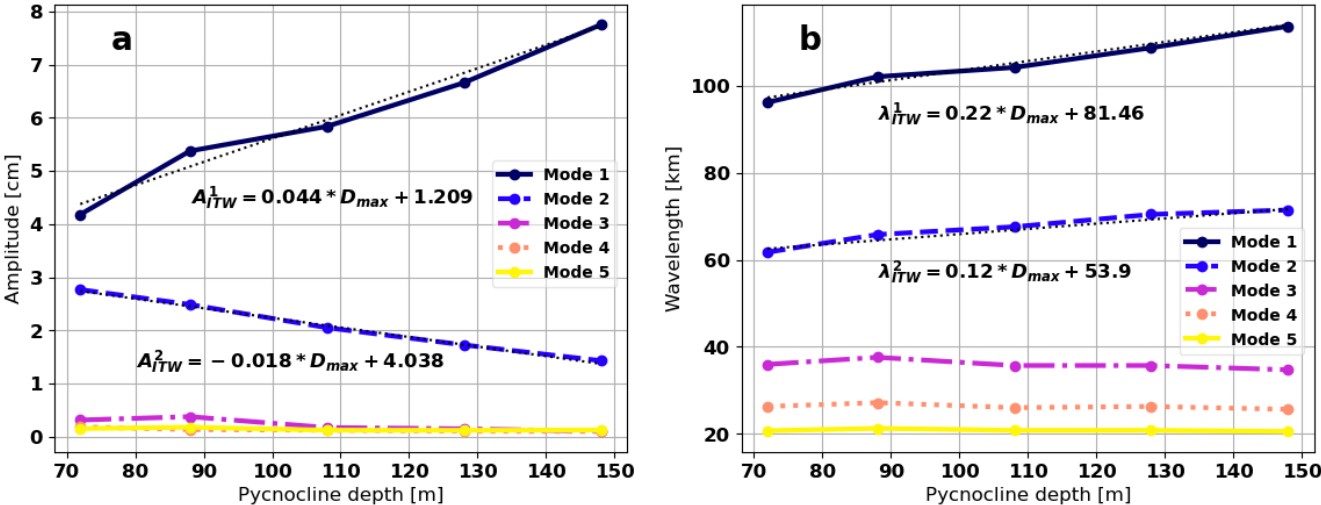

**Figure 7.** (a) Amplitude and (b) wavelength of the surface elevation for each vertical mode with respect to the pycnocline depth. The calculations are based on the Amazon shelf simulations over the abyssal domain ($-300$km$< x < 200$km). The fit equations for the two first modes are shown below the curves (with $A_{\text{ITW}}^n$ the ITs' surface amplitude for mode $n$, $\lambda_{\text{ITW}}^n$ the ITs' surface wavelength for the mode $n$ and $D_{\text{max}}$ the pycnocline depth).

The number of troughs for mode 2 decreases with a deeper pycnocline. The wavelength of the combination of all modes seems to be larger with a deeper pycnocline.

To investigate this impact, the wavelength of each mode has been calculated for the abyssal domain (from $-300$km to 200km). The wavelength is calculated following the method of Welch (1967) with 200 elements per segment and a zero padding of 5000 elements. Using the zero padding enables to increase the resolution of larger wavelengths but create irrelevant wavelengths, so all the wavelengths larger than 500km are cleared ($\lambda_{\text{max}} = Ndx$).

    Figure 7 details the evolution of the ITs amplitudes and wavelengths at the surface with respect to the pycnocline depth.

As expected with the shape of the modal structures, modes 1 and 2 are linearly controlled by the pycnocline depth. A deep pycnocline increases the wavelength of the mode 1 and 2 of the surface elevation with a slightly stronger impact on mode 1. The amplitudes and wavelengths of modes 3, 4 and 5 seem to slightly decrease with the increasing depth of the pycnocline. Empirical relations for modes 1 and 2 can be found by fitting the curves.

    These relations could be a useful proxy to determine the ITs' surface wavelengths but have to be used with extreme caution.

As explained in the introduction, the ITs' surface wavelength is directly dependent on the bathymetry. In this simulation the bathymetry is always the same, allowing us to formulate this empirical relation. Hence, this relation is only valid in the 4000m deep tropical areas studied here. Other tests should be made in order to integrate depth variations into these empirical relations.

    This variation of the ITs' surface wavelength can lead to a strong aliasing of the ITs corrections and altimetric observations if this variability is not taken into account. For example, the wavelength difference for the mode 2 between AS-71m and AS-





148m is about 10km. With only three occurrences of the wave beam at the surface, the shift associated to the mode 2 is about 30km. The correction that only use a wavelength of 60km to corrected mode 2 (corresponding to AS-71m), will be in phase opposition after only three occurrences if the stratification lead to an actual wavelength of 70km (corresponding to AS-148m). This rough calculation helps to understand that small changes in the wavelength can completely change the shape of the surface ITs signature.

Tchilibou et al. (2019) reported a similar observation comparing two simulations from El Niño and La Niña context in the Salomon Sea. In this region, the El Niño stratification is characterized by a shallow pycnocline and the La Niña stratification is characterized by a deep pycnocline. The authors pointed out that this stratification variability is one of the sources of the non-stationarity of the ITs observed using altimetric SSH. The surface elevation of the model outputs seems to present a larger wave beam during La Niña than during El Niño (Tchilibou et al., 2019, Figure 8). In the Amazon shelf area, the high dependency of

the wavelength to the pycnocline depth could be a major contribution to non-stationary ITs that appear in the study of Zaron and Ray (2017).

This result reinforces the importance, in the future, to properly take into account the pycnocline depth regimes in order to estimate a more accurate ITs' surface signature. The pycnocline depth could be used as a proxy to estimate the wavelength of ITs and then, better constrain the ITs atlases. The simulations of the different clusters should be done for a broad range of max-

imum bathymetry. This work could enhance the empirical relations of ITs amplitude and wavelength adding the dependency to the bathymetry.

To compare the T-UGOm simulations to realistic studies, some atlases of ITs and a realistic NEMO simulation are used. In Ray and Zaron (2016), the ITs' surface elevation amplitude is around 6cm. In Zaron (2019), the ITs' surface elevation amplitude is around 4cm. This difference can be first explained because long term altimetry harmonic analysis only extracts

the stationary part of the ITs (as explained in the introduction). So these empirical atlases only resolve the mean stratification context of the ITs. Over the year 2015 in the NEMO simulation of Ruault et al. (2020), the ITs' surface elevation amplitude is around 5cm (publication in preparation). With an amplitude ranging between 7cm and 10cm, the simulated amplitudes are higher compared to those altimetry-derived atlases and realistic simulation but are the right order of magnitude.

In Table 1, AS-1, AS-2 and AS-3 (AS-72m, AS-108m and AS-88m) represent more than 70% of the density profiles. These

clusters only represent profiles with a shallower pycnocline. AS-108m is the closest cluster to the median of all the profiles, this cluster is used for the comparison with realistic products. The ITs' surface elevation amplitude of AS-108m is 7.5cm, closer to the altimetry atlases than the 10cm of AS-148m. However, the intensity of the ITs are also strongly influenced by the intensity of the barotropic tide energy flux directed off the shelf slope. The barotropic currents are mostly set through the barotropic boundary conditions (see Tab. 2 for the barotropic currents used in the present study). To some extent, only the relative changes

of ITs characteristics should be considered quantitatively when comparing our academic experiments to empirical analysis or 3D realistic simulations.





### 3.2.2 Impacts of the Bay of Biscay stratifications

In this section, the clusters are sorted following the period of the year they represent: BB-1,3,6,4,5,2 To facilitate the description, the clusters are renamed with the corresponding season: BB-winter, shoulder, spring, summer, hot-event, fall.

In the Bay of Biscay case, the differences between the typical profiles are not driven by the surface pycnocline depth but by the stratification value in the upper surface layer (<25m, Fig.4b). Indeed, the maximum value of $N$ has a great variability whereas the pycnocline depth is always close to 40m (Tab. 1) except in winter where there is no surface stratification (BB-1 in the table). Because of the $N$ variability, the interpretation of the stratification impact on the baroclinic modes (Fig. 8 left panels) is more complex than for the Amazon shelf simulations.

The easiest way is to proceed from lower modes to higher modes. Mode 1 is almost the same for all the clusters for both vertical ($w$ and $\eta$) and horizontal structures ($u,v$ and $P$). So mode 1 is not sensitive to the variability of the stratification. Indeed, mode 1 is built upon the constant maximum of the stratification around 800m (Fig. 2f).

Mode 2 presents the same trend for both vertical and horizontal structures: the more stratified the cluster is, the more the mode is shifted toward the surface. The order of the trend sorts the clusters as follows: BB-shoulder, BB-spring, BB-fall, BB-

summer, BB-hot-event. BB-winter is excluded from this trend. For mode 2 horizontal structures ($u$, $v$ and $p$), BB-winter and BB-shoulder are exactly the same at the surface and BB-winter is stronger at intermediate layers. This can be due to the fact that the only stratification of the BB-winter profile is located around 800m. For mode 2 vertical structures ($w$ and $\eta$), BB-winter is stronger than BB-shoulder at the surface. No explanation could be found for this pattern.

For mode 3 vertical structures, the trend at the surface is the opposite of mode 2: BB-hot-event, BB-summer, BB-fall,

BB-spring. BB-winter and BB-shoulder are excluded because they are weaker than BB-hot-event. For intermediate layers BB-shoulder fits to the trend. For mode 3 horizontal layers at the surface, the previous trend is completely irrelevant. The clusters are classified as follows: BB-winter, BB-shoulder, BB-hot-event, BB-fall, BB-summer, BB-spring. This could be due the stratification at the surface, as the density profiles seem very straight at the surface in Figure 4d.

The overall baroclinic amplitude of the surface elevation ranges between 3cm and 8cm (Fig. 8 right panels). Modes 1

and 2 are stronger with the increase in stratification between 20m and 60m: being 3cm−0.5cm for BB-winter and 3.8cm-3cm in the highly stratified BB-hot-event. Mode 2 is more sensitive to the stratification: it is almost equivalent to mode 1 for BB-hot-event whereas it is almost null for BB-winter. Mode 3 is stronger dunring BB-shoulder, BB-spring and BB-fall than for other stratification where it is almost null. This confirms the observations made for the modal structures. Modes 4 and 5 are only visible during BB-shoulder. These figures also highlight the fact that ITs are completely absent from the shelf

domain ($x > 250$km) during BB-winter, when the stratification between the surface and 200m is null and does not support the propagation of ITs.

In summary, in the Bay of Biscay, mode 1 is controlled by the maximum of $N$ at 800m, mode 2 is controlled by the value of $N$ between 20m and 40m and mode 3 might be controlled by the value of $N$ between the surface and 20m.

The T-UGOm simulations are also compared to the same atlases as for the Amazon shelf. In Ray and Zaron (2016), the ITs'

surface elevation amplitude is around 1cm. In Zaron (2019), the ITs' surface elevation amplitude is around 2cm. The simulated



**Figure 8.** Baroclinic modal structures for the three first modes (left panels) and simulated amplitude of baroclinic surface elevation for the five first modes (right panel) for all the typical density profiles of the Bay of Biscay. The modal structures are different for vertical processes ($w$ and $\eta$, upper left panels) and horizontal ones ($u$,$v$ and $P$, lower left panels). The black points show the extremums of the modes. The simulations are sorted with respect to the seasons. On the right panels, the white line represents the sum of the baroclinic modes, the colored patches represent the part of each mode in the sum: the modes on top of the sum line refer to destructive interaction between the modes.





amplitudes between 3cm and 8cm are higher compared to those altimetric atlases and simulation but are the right order of magnitude. The difference could come from the difference in the barotropic tidal forcing and the bathymetric slope that are arbitrary set in these idealized simulations. But as explained in the introduction, the empirical models of ITs also probably under-estimate the amplitude of the ITs' SSH because of the large non-stationary component in this region. Also, the altimetric

observations are not suitable to capture the surface signature of the higher vertical modes because their wavelengths are too small compared to the spatial distribution of the observations. So the results of Ray and Zaron (2016); Zaron (2019) mostly highlight the relative stationarity of the first mode, and are not showing the weaker ITs amplitudes from the higher modes.

The number of peaks of mode 2 occuring in the domain from $-300$km to $200$km give an approximation on the wavelength sensitivity in the Bay of Biscay: there are 6 peaks of mode 2 for BB-winter, 5 for BB-shoulder, 6 for BB-spring, 6 for BB-

summer, 5 for BB-hot-events and 5 for BB-fall. The difference is clearly visible between BB-summer and BB-hot-event at $-50$km: the surface amplitude combination of the modes is minimum for BB-summer and maximum for BB-hot-event. This suggests that the stratification contributes to the variability of the wavelengths of the ITs modes in the Bay of Biscay as well.

As previously explained, the surface pycnocline depth is not the right parameter representing the ITs variability in this area. The values of the stratification at different depths have been explored but the wavelengths are not influenced the same way as

the amplitudes are. As no adequate proxy has been actually found (such as the pycnocline depth for the Amazon shelf), both the ITs' surface amplitudes and wavelengths are processed seasonally. A climatology has been constructed with a time step of 3 days. For each period of 3 days, the amplitudes and wavelengths of each mode are calculated from the weighted mean of the clusters distribution in the dataset. The weighted standard deviation is also calculated to evaluate the variability inside each time step.

Figure 9 shows this climatology. As expected, the amplitudes and the wavelengths do not present the same pattern. To simplify the descriptions, BB-winter characteristics are used as a reference for the comparisons. The amplitudes of each modes are very contrasted through the year. Mode 1 is stronger from August to October with a homogeneous value of 3.5cm through this period. It became weaker in May (around 2.7cm), and maintains a plateau of 3cm from December to May. Mode 2 has larger amplitude variability compared to other modes. From 0.5cm in winter and spring, it increases to 2.5cm in September,

then decreases until January. Even with smaller amplitude, mode 3 has significant amplitude variability: two peaks at 1.3cm and 1.0cm happen in June and November, otherwise stabilized around 0.3cm. This particular pattern seams to be due to the stratification near the surface, stronger in BB-shoulder and BB-spring compared to other cluster that are well mixed near the surface. Modes 4 and 5 amplitudes patterns are close to mode 3 one but the peaks happen in May and December, so mainly caused by BB-shoulder surface stratification.

The wavelengths have a weak variability between all the simulations. This explains why the standard deviation is almost null. The wavelength of mode 1 is almost constant during the year, it only decreases in May and December, due to BB-shoulder stratification. The wavelength of modes 2, 3,4 and 5 have similar pattern around two values: lower from February to May (50km, 35km, 26km and 20km) and stronger from June to December (56km, 45km, 33km and 25km). The shifts are stronger for modes 3 and 4 and is clearly due to the presence of surface stratification after winter. In Figure 8, the clostest extremum to

the surface of the modal structure of horizontal structure $(u, v, P)$ is affected by the stratification and is not located at the same





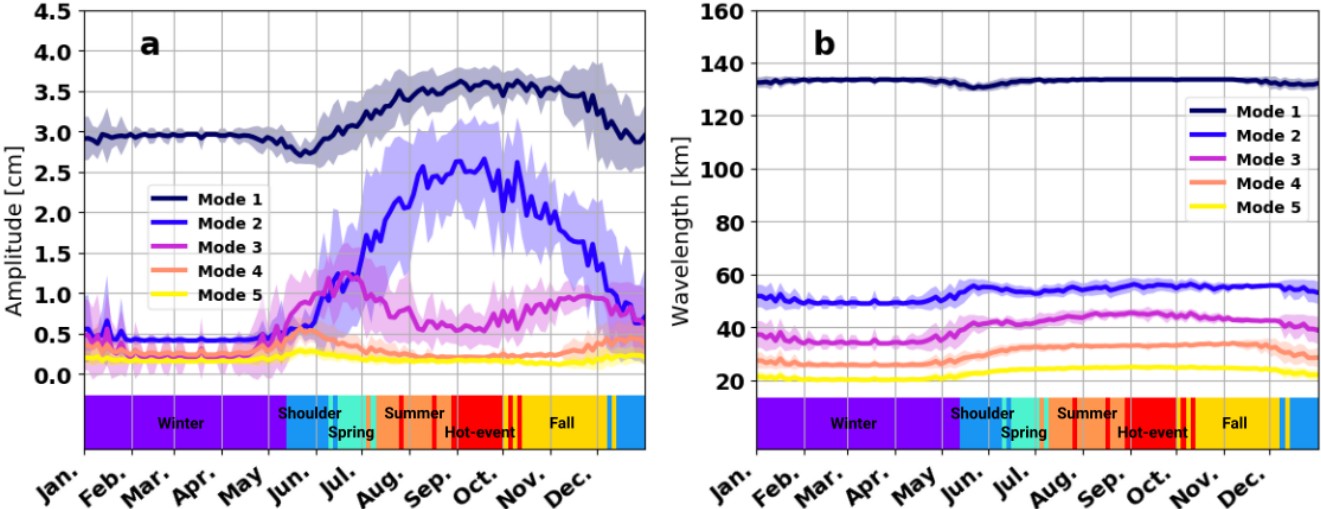

**Figure 9.** Weighted mean of (a) the amplitude and (b) the wavelength of the surface elevation for each vertical mode during the year. The calculations are based on the Bay of Biscay simulations over the abyssal domain ($-300$km$< x < 200$km). The climatology is built with a time step of 3 days where the ratio of each cluster is used as the weight. The shading represents the weighted standard deviation. The ruler at the bottom of the plots shows the color of the dominant cluster along the year.

depth. For mode 2, the variability of the depth is around 400m but for mode 3, the variability is around 1200m. This could explain why the wavelength of mode 3 is more affected than the wavelength of mode 2. Thus, the vertical modal structure and their variability based on the stratification can give a good estimation of the wavelength response.

As with the Amazon shelf, these variations in wavelength can be responsible for significant shifts in the surface elevation
patterns and lead to wrong corrections of the ITs' surface signatures. Still, the relatively weak variability in wavelengths might explain why the non-stationarity of the ITs is weak in this area in the maps of Zaron and Ray (2017). This cluster-based climatology is the first step to build an evolving correction of the ITs during the year. This climatology could help to better estimate the amplitude observed from altimetric observations.

### 3.3 Discussions

The ITs simulation experiments clearly highlight the dependency of the ITs patterns of amplitude and wavelengths (hence surface signature) on the stratification. They provide a quantitative estimate of the ITs temporal variability due to the ocean background stratification changes only. This background stratification is a key process for the ITs generation and one of the processes that control the ITs propagation.

As the focus of this study was the background stratification, other parameters have been set equally in both configurations
(Amazon shelf and Bay of Biscay). The slope and the barotropic tidal forcing are two key parameters for the ITs generation. These parameters, combined with the background stratification, directly control the ITs generation and with it the amplitude





of the surface elevation. This is why the present study is focused on the inter-comparison of background stratifications rather than the realistic values of ITs. This enables us to compare the different stratification pattern between the two areas of interest and conclude that the pycnocline depth (Amazon shelf) leads to a stronger impact over ITs surface signature than the surface
stratification (Bay of Biscay)

The slope of the shelf is a bit steeper in the Bay of Biscay than in the Amazon. Based on FES-2015b (Lyard et al., 2021), the M2 tidal barotropic currents in front of the Amazon shelf are around $3\mathrm{cm.s}^{-1}$ and in front of the shelf of the Bay of Biscay are around $5\mathrm{cm.s}^{-1}$. The forcing used for the COMODO test case was $10\mathrm{cm.s}^{-1}$ which is much stronger than the realistic forcing. This could explain why the simulated amplitude are stronger than the one of the atlases in both cases. Using the realistic slope
and tidal forcing could also reduce the difference of amplitude between the two areas. Indeed, the tidal forcing is stronger in the Bay of Biscay but the background stratification is stronger in the Amazon shelf.

To further explore the comparison with realistic products, the ITs' surface wavelength is also extracted from the atlas HRET V8.1 (Zaron, 2019) and from the NEMO simulation (Ruault et al., 2020). The comparison is only made for the Amazon shelf. The wavelength calculation is made with a 2D Fast Fourier Transform of the ITs' surface elevation complex field. Using the
complex field enables us to extract the direction of propagation in addition to the wavelengths. Because the NEMO simulation only cover the area 2°S-9°N, 52°W-43°W, the same area is used for HRET.

Figure 10 first shows that the wavelength pattern is very similar between HRET and NEMO. They highlight a major wavelength of 110-120km propagating northeastward and secondary wavelengths from 70-80km propagating in the same direction. In NEMO, the propagation of this secondary wavelength is stronger and wider than in HRET. This could be due to the con-
straints used in HRET to better structure the ITs. The phase field of HRET shows ITs propagation with wave crests roughly linear whereas the NEMO phase field shows them more smoothly curved (Fig.10c,d).

These wavelengths are slightly longer than the one calculated for AS-108m. As explained in the introduction, the ITs' surface wavelength is directly dependent (roughly linearly) on the bathymetry. The bathymetry of the T-UGOm simulations is set capped to 4000m whereas the real bathymetry in the area can extend down to 4500m in the generation zone and down to 5000m
further north. These bathymetry differences likely explain almost all of the wavelengths differences. Still, the stratification variability could also contribute to these differences. The ITs' surface wavelength of AS-148m is around 115km for mode 1 and 70km for mode 2. This fit the wavelengths of HRET and NEMO almost perfectly.

More realistic experiments with various uniform stratifications in a realistic regional grid are presently carried out for the ITs in the Bay of Biscay with the model SYMPHONY (Marsaleix et al., 2008). So far, these simulations lead to similar conclusions
compared to the ones obtained with the T-UGOm academic configuration (Barbot, 2021, PhD thesis, in preparation). This confirms that the T-UGOm simulations, although very academic, are in the right range compared to more realistic cases. Such idealized simulations are thus a good way to estimate the ITs properties before running more realistic but complex 3D simulations.

The idealised approach could also help to furnish a simple correction from the background stratification in order to better
understand the complex dynamical interactions between the ITs and the mesoscale oceanic circulation. This correction could




**Figure 10.** Two dimensional Fast Fourier Transform of the baroclinic surface elevation for M2 from (a) HRET V8.1 atlas (Zaron, 2019) and (b) a realistic regional NEMO simulation for 2015 (Ruault et al., 2020) and (c,d) their associated phase field. The grey rings represent the wavelength grid scale and the black rings represent the wavelength of the T-UGOm simulation for AS-108m. The bold wavelength is calculated from the overall elevation whereas the others are calculated from the two first vertical modes.





be enhance using the clustering methods with these a priori cluster to link the density profiles to the closest cluster. Once the density field is classified, the corrections from the corresponding idealised case could be applied.

## 4    Conclusions

The classification of density profiles through clustering methods can be very useful to describe both spatial and temporal
variability of the stratification. As shown, this methodology can highlight different regimes of stratification that are linked to seasonality (Bay of Biscay) or to spatial distribution (Amazon shelf) at the same time. Thus, any kind of stratification variability can be handled with a single methodology. Especially for cases that are not driven by seasonality or for cases with clear spatial distribution variability, this methodology is a great improvement compared to mean state and seasonal classification.

The users of such a cluster methodology need to be aware of some specific parameters. The first and more important one is
the normalization of the profiles. This choice is important and can change the goal of the classification. The second point is the choice of the clustering method for systematic or automated stratification studies. Many of clustering methods exist with different performances, but a first selection can be made by looking at the distribution of the PCA manifold.

For the Amazon shelf, the clusters help to highlight the strong spatial variability of the stratification, and the dominance of the pycnocline depth in this variability. In the Bay of Biscay, the cluster does reproduce the seasonality of the stratification and
highlights two different regimes for the summer.

The present results of ITs simulations allow a better understanding of the ITs dependency on the background stratification. This dependency not only occurs in areas driven by the radiative forcing but also in areas driven by the circulation. First, the stratification variability has a stronger impact on ITs if the stratification is composed of only one pycnocline. In the tropics, such a pycnocline is maintained during the year and is stronger than the ones in mid-latitudes. Second, the pycnocline depth
has stronger impacts on both amplitudes and wavelengths of the two first modes of the ITs than the surface stratification. In the presence of a strong ocean circulation, the variability of the pycnocline depth is more important. Third, the surface stratification variability, driven by the radiative forcing has a stronger impact on the amplitudes of modes 2 and 3.

The high dependency of the ITs horizontal wavelengths, amplitudes and modal distribution to the stratification regimes is a major result of this study and also leads to a better understanding of non-stationarity of the ITs. This result will impact on the
future works dedicated to ITs' surface elevation observation and prediction. Temporal harmonic analysis of the surface elevation can only estimate the ITs anomalies of the stationary part, dominated by mode 1. The present study highlights that mode 2 can be nearly as strong as mode 1 in both areas of interest. This result suggests that time harmonic analysis underestimates the ITs multi-mode amplitude and omits the ITs wavelength of mode 2. Moreover, the ITs' surface elevation corrections based on such methods, without realistic horizontal wavelengths, could create a fictitious signal in the corrected observations from the
aliasing between the real ITs wavelengths and the wavelength of the correction. The frequency domain modeling proposed in this study could be used to build multiple simulations with various stratification regimes that could then serve as references or constraints for ITs corrections atlases. This approach should be used preferably for regions where the stratification regime can set, as in the mid-latitudes areas with weak circulation. However, for regions with highly variable stratification regimes and
strong circulation, this approach should be used with caution. Such modeling would not be representative of the circulation
and also could be highly unstable.

Finally, coupling the two approaches of clustering methods and the academic simulations results in the production of two types of climatologies of the ITs amplitudes and wavelengths for the five first baroclinic modes. For the Amazon shelf, the wavelengths can be linearly derived from the pycnocline depth for a constant bathymetry. Whereas in the Bay of Biscay, the wavelengths have to be determined progressively during the year. The clustering methods enable us to set the delimitation of
the seasons based on the stratification rather than on monthly climatologies. The definition of a good parameter controlling the ITs amplitude and wavelengths need to be pursued in mid-latitude to unify the processing of the different regions of the global ocean.

Currently, the SWOT mission encourages international efforts in order to separate the mesoscale and ITs' surface elevation contributions. This study invites other researchers to carefully consider the background stratification and even more its vari-
ability within the different approaches in usage by the community, in order to predict and remove the ITs' surface elevation signature.

*Code and data availability.* The hydrodynamic code T-UGOm (CECILL licence) is available at https://hg.legos.obs-mip.fr/tugo Mercurial repository. The climatologies of the ITWs properties in the Bay of Biscay are available upon demande to the author.

*Author contributions.* SB developed the clustering methodology, runned the simulations and realised all the graphics and interpretations. FL
developed T-UGOm, built the model configuration for previous studies and supervised the runs. FL, MT and LC enhanced the interpretations and graphics with their relevent advices.

*Competing interests.* No competing interest to declare.

*Acknowledgements.* The authors are grateful to the CNES, CLS, CNRS and UPS for the funding of this study as well as the projects COCTO (PIs: N. Ayoub and P. De Mey-Frémaux) and HighFreq (PI: F. Lyard) for the additional funding of diverse missions for the communication of
this study. D. Allain is also thanked for his help for the simulations and the development of POCViP. The developers of Scikit-learn Python package (Pedregosa et al., 2011) are thanked for providing a well-documented package on machine learning algorithms. Rosemary Morrow is thanked for her helpful comments about the smooth understanding and the writing of this article.





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



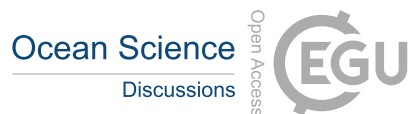

## Appendix A: Amazon shelf additional visualisation of the clusters

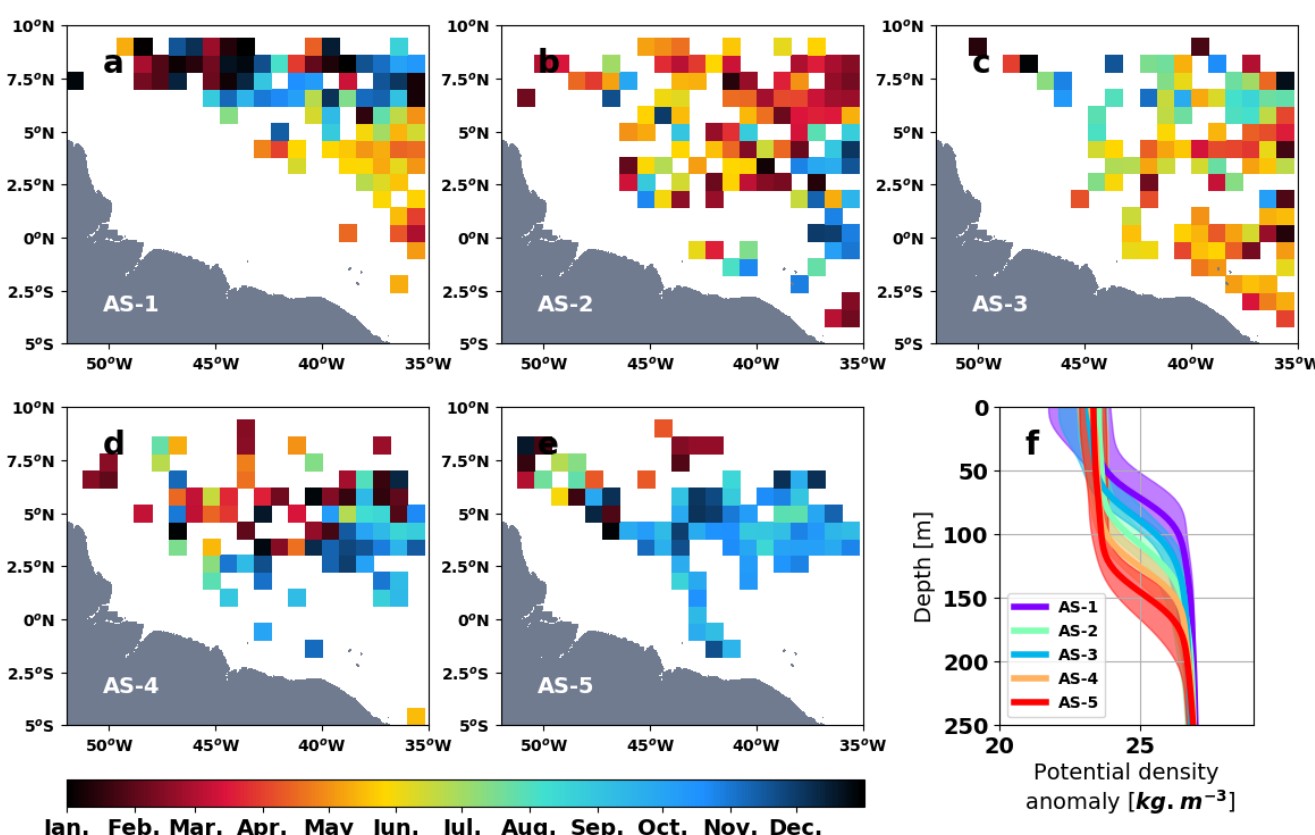

**Figure A1.** (a-e) Mean seasonality over the area for each cluster in the Amazon shelf and (f) the median profiles of the clusters from the figure 3. The mean season is processed over the area within boxes of 1x1 degree, only the boxes with more than 2 profiles are showed.