# Peer review of "The dependency of internal tides on background stratification variability: a case study on the Amazon shelf and the Bay of Biscay"

_Ocean Science, 2021_

## Author Response (AR1)

We do thank Referee#1 for his/her careful reading of our manuscript and relevant comments. Below his/her comments are listed (after #, in italic) and followed by our answers (after >>) as well as the modification in the manuscript (between "..."). Changes made in the manuscript are highlighted in red and blue in the revised version of the manuscript. As some remarks of the General comments are also addressed in more details in the Specific comments, we took the liberty to reorganize the document by gathering together the comments tackling the same issue, in order to make our answers clearer. Each subject of the general comments is gathered under a dedicated title. We hope this will ease the reviewer's reading.

**Overall comment #1:**

*In that last case (western equatorial Atlantic), I would also emphasize that all modes are not necessarily present in the same areas (for example in one season mode 1 has a strong dominance in northern NBC retroflection, and thus not present near the shelf break closer to the equator, and modes 2 and 3 on the other hand seem to be more present near the equator. Then, between 2 and 5°N, there is a bit o AS-1 and AS-4, AS-5. I mention that, as I think that the whole range of solution explored (for example in figure 6, right panels) might not necessarily be present at the same place (at different times). Thus, maybe proceding in that way might locally overestimate the range of variability in ITs characteristics that is possible due to changes in stratification.*

>>The spatial distribution of the clusters in the western equatorial Atlantic is exposed with Fig.3c and mention l.303. In order to clarify the discussion, one general sentence in the results (l.297) and a dedicated paragraph at the beginning of the discussion (l.364) are added:

l.297: "In addition, the spatial distribution of the cluster is not homogeneous within the area highlighting spatially-bound processes responsible for some specific stratification."

l.364: "The stratification of the two areas of interest are driven by very different forcing: the Amazon plume and the circulation for the western equatorial Atlantic and the radiative forcing for the Bay of Biscay. The Amazon water recirculation at the North of the domain and the NBC rings along the shelf break are limited-extent processes whereas the radiative forcing affects the Bay of Biscay homogeneously. Thus, the spatial variability is stronger in the western equatorial Atlantic. In such area, the stratification variability presented by all the clusters does not happen in every parts of the domain but this method enables us to distinguish the specificity of each sub-region at once."

**Overall comment #2:**

*#In the presentation, I also wondered about the choice of presenting together the clustering in the two regions, and then the TUGO solutions in the two regions. I wonder whether it would not make more sense to have one chapter for the western equatorial Atlantic (clusters, then simulations), followed by one chapter for the Bay of Biscay (cluster, then simulations).*

>>This choice is made in order to emphasize the variety of stratification variability that the clustering is able to handle with a common setup. Addition at the end of the introduction:

l.86: "This organization enables to better compare how the presented methodology can handle the two areas that have two different stratification variabilities."

**Overall comment #3:**

*#I also found the paper's title misleading. There is no discussion in it of internal tides on the Amazon shelf. The AMazon shelf is not really considered, the profiles analyzed are not on the shelf (where there is often a large freshwater cap), and the model solutions discussed are for the deep ocean ot its east. I would thus strongly recommend a title change. Maybe: '... : case studies for the western equatorial Atlantic and the Bay of Biscay'*

**Related comments**:

*#l. 115-119, this is not a very good introduction for the Amazon shelf... (where freshwater inputs are major sources of stratification variability) (I realized after that most of the emphasis and investigation is not on the Amazon shelf, but on the nearby deep tropical Ocean; for that, the title should not be 'a case study on the Amazon shelf...).*

*#l; 153: from what I know, the latest versions (>5.0) also include the individual profiles. I would use one of these versions which also includes non-Argo data, and thus CTD data from cruises on the shelves (that are clearly mostly absent of the data set used for the analysis presented) (later I read (page 10) that the period investigated was 1984-2015; this period includes a large number of research cruises and surveys on both shelves).*

*#Figure 2: only show data that are not on the shelves (?)*

*#l. 175: 'For the Amazon shelf area...' misleading. The figure clearly shows that the Amazon shelf is almost ignored (almost no data). Similar in the Bay of Biscay for the shelf areas. These are regions with lots of CTD cruises (in particular Bay of Biscay, but even on the Amazon shelf, there is a large number of CTD (and Ocean stations) available, as can be seen by a quick search with the NCEI tools, but I am sure that the same would be true for recent versions of CORA (version number > 5.0)*

*#l. 188: does I understand correctly that profiles not extending to at least 600m (Amazon shelf) and 300m (Bay of Biscay) are not considered (thus removing all shelf areas, as well as part of the slope areas).*

*#l. 356: do not refer to the data set as 'Amazon shelf'.*

>>Thank you for pointing that out.. We used the term "Amazon shelf" because we cannot find a proper name for this part of the ocean, but we realize that it can be confusing as the focus is not made on the shelf itself. Indeed, the term you propose "western equatorial Atlantic should be less confusing. It has been modified everywhere in the article. "AS-#" clusters are renamed "WEA-#". A new title is proposed:

"Background stratification impacts on internal tides generation and abyssal propagation in the Western Equatorial Atlantic and the Bay of Biscay "

CORA V4.3 does provide the cruises data and some of them are used in the selected data we used.

**Important comments :**

*#l. 79: 'stratification is not a force'. I think that the first sentence should be reworded. Maybe 'stratification is what 'controls' the internal waves'.*

>>Reformulated:

"The stratification controls the buoyancy of the water masses that is the restoring force of the internal waves."

**l. 95: notice that the dispersion relation is 'local', and not as the sentence starts with 'For a given N profile'**

>>Thanks you for pointing that out. "For a given N is removed".

**Figure 2: What is the unit of the number of profiles (panel d) and what is the period considered for the profiles.**

>>The number of profile has no units: " Number of profiles [#]".  Period used: 1984-2015. Figure edited.

**l. 190-192: these lines of comments should be placed elsewhere, probably in the introduction.**

>>Moved to the section describing the model configuration (l.436-439).

**l. 205: what are these two PCA axes? Why two?**

>>Addition of few sentences to clarify this point:

l.220: "As the profiles are only described by the density versus depth, only two principal components are used. Thus, the shape of each profile is evaluated according to two orthogonal axes. The PCA manifold is the plan defined by these two new axes and where each profile is characterized by a point. The both axes explain a different part of the standard deviation of the profiles. For example, if the profiles are mainly controlled by the pycnocline depth but also by the surface density, the first axis (PC1) will be controlled by the different depths of the pycnocline whereas the second axis (PC2) will be controlled by the surface density."

**l. 115-119,  For the surface layer, instead of circulation, I would indicate 'water masses'.**

>>The water masses are already mentioned: l.131 "The second one is the circulation of the ocean and the induced mixing of the water masses." Maybe it is better this way:

"The second one is the mixing of water masses induced by the circulation."

**l. 236: I don't understand the sentence 'For the Bay of Biscay, 10 clusters are not enough...( why does it imply that 6 clusters is a good compromise, as mentioned in the next sentence).**

>>The section 2.2.2 (now 3.2.2) have been rewritten on the advice of Referee #2 and further details are presented about the clustering methods tests in the appendix A. This specific sentence is reformulated:

l.249: "For the Bay of Biscay, the variability of the density profiles is way more complex and N profiles are very different even for 10 clusters. But a high number of clusters leads to have some clusters with few profiles. Thus, for both areas, a classification of 6 clusters is a good compromise that enables us to detail the evolution of the density profiles while keeping well represented clusters (more than 100 profiles)."

**l. 244: I don't understand what is done. The previous analysis done to 600m (300m) for the two regions. Is it that the vertical profiles are then considered below those depth ranges, and a median average is estimated for the profiles included in each cluster?**

**l. 247: 'as such a deep depth'? Which deep depth is considered here? (is it 600m or 4000m? anything in between?)**

>>The model configuration is defined from 0 to 4000 m depth so the density profiles used for the modeling need to be extended. This paragraph and the one before are reformulated:

"Once the classification is done, the typical profile from each cluster is calculated in order to use it as a forcing in the simulations (see section 3.1). The density profiles need to be strictly stable and defined from 0 to 4000 m depth.

Because most of the profiles used for the classification are not defined that deep, the completion process is detailed here. The median of the profiles within the clusters is used as deep as possible. The standard deviation below 1000 m is very weak (Fig. 2a,f) so the profile can be completed with the median of the profiles from the other clusters. If the profile does not reach 4000 m, then the bottom of the profile is extrapolated using the density gradient of the latest 4 measurements. The density gradient used for the extrapolation needs to be weaker than $5.0*10^{-7}$ kg.m$^{-4}$ that is a common gradient at such a deep depth.

The median profiles have been smoothed …"

*#l. 289: similarity between AS-2 and AS-3. This is interesting, but why did the analysis project the profiles into the same category. I am also confused in this paragraph, which starts that the NBC is strongly influenced by large anticyclonic eddies, and ending by the clusters identify… the steady state of the NBC.*

>>Mention added that profiles in WEA-3 have a shallower pycnocline that the ones in WEA-2. This paragraph is reformulated:

"WEA-2 and WEA-3 are similar: they have the same seasonality, the same spatial coverage and gather the profiles with a pycnocline depth from 80m (WEA-3) to 110m (WEA-2). The North Brazil Current (NBC) is a strong geostrophic current flowing along the Amazon shelf break all year-round. The seasonality of the NBC is mainly influenced by wind-driven eddies from August to November that enhance the retroflection of the NBC water masses into the North Equatorial Counter Current (NECC, Johns et al. 1998). The seasonality of WEA-2 and WEA-3 as the large spatial distribution of the clusters clearly point out that they identify the steady state of the NBC, without the eddies."

*# I guess that the separation between clusters 2, 3, 4, and 5 is a rather continuous transition in pycnocline depth. I was wondering whether maximum $N^2$ also changes between the cluster, but at first glance this does seem to be the case on figure 3. Another puzzler is that surface density of AS-1 does not seem particularly less than for the other cluster (if I read well figure 3). I would have expected lower surface density as it includes the Amazon plume in NBC retroflection (but also other profiles in the other seasons south of 5°N).*

>>The table 1 does highlight that the $N^2$ value is the same for all the clusters ($2,3-2,5*10^{-2}$s$^{-1}$). The 90% interval of the cluster 1 highlights more low surface density profile than for the other cluster, but yes, the median of the cluster 1 is the same as the other clusters. Maybe it can be explained because the profiles are quite far from the shelf break so the Amazon waters have already mixed down to 50 m.

*#l. 346: I am not sure that the differences between the two ranges of years included in ISAS13 and CORA V4.3 explains the difference. I checked in recent years, and the CORA stratification remains. If anything, using a shorter period sharpens it? I suspect that spatial (and even more) vertical smoothing could influence the weaker vertical gradients in ISAS13.*

>>The sentence is reworded and the following comment added:

"ISAS13 climatology is only based on the 2004-2014 period but this different period cannot explain all the differences either. The spatial and vertical smoothing applied to construct the climatology might have been stronger in this area compared to the filtering we used here."

**l. 376: what is meant by 'There, the cluster classification is more concise'?**

>>Reformulated:

"The 6 clusters classification gathers the same amount of information about the seasonality as the 12 groups of the monthly classification. Thus, the cluster classification is a more condensed approach."

**l. 529: 'the trend' should be replaced (in most cases in this section) by 'the change..'**

>>Is the term "pattern" better than "change" ?

**l.601: 'the slope of the shelf'. I am wondering whether it is shelf or slope region that is considered (I believe the 'latter'). At the end of the line 'than in the Amazon' ? What is meant there.**

>>Reformulated:

"The slope of the shelf break is a bit steeper in the Bay of Biscay than in the Amazonian region"

**l. 634: 'As shown here, this approach is very useful...'**

>>Can the conclusion start this way ? Reformulation:

"The classification of density profiles through clustering methods is very useful to describe both spatial and temporal variability of the stratification."

**Minor Comments:**

**L. 84: 'Those wavenumbers project on different vertical modes.' (I am not sure I would mention the first mode being the barotropic mode, as the vertical wavenumber mentioned will not project on the barotropic mode).**

>>Right, the bracket is removed.

**l. 113: 'Before investigating these impacts, we will discuss the range of stratification variability that needs to be investigated'**

>>Sentence replaced.

**l. 170: I would remove the sentence 'The ITs induce pressure oscillations...' (not just the Its, but any adiabatic vertical motion)**

>>Sentence removed.

**l. 263-264: erroneous Argo float profiles are found by this approach. This means that there was an error in the flagging. This is quite possible in version 4.3, but should not have happened in more recent versions, where a test of 'possible min-max range is applied at each depth of each profile.**

>>Duly noted.

**l. 394 – 396: these sentences can be removed. Maybe to be mentioned in discussion or conclusion.**

>>Removed.

*#Fig. 6: I don't fully understand what is represented on the right panels. Is it surface elevation? I don't see contributions of modes 3 to 5. Is it that they are negligible, as suggested by Fig. 7 (in which case, no need to put them in caption).*

>>Yes it is the surface elevation. The modes 3 to 5 are negligible so they are removed from the figure.

All other minor comments are accepted as formulated by the Referee #1.

We do thank Referee#2 for his/her careful reading of our manuscript and relevant comments. Below his/her comments are listed (after #, in italic) and followed by our answers (after >>) as well as the modification in the manuscript (between "..."). Changes made in the manuscript are highlighted in red ad blue in the revised version of the manuscript. As some remarks of the General comments are also addressed in more details in the Specific comments, we took the liberty to reorganize the document by gathering together the comments tackling the same issue, in order to make our answers clearer. Each subject of the general and specific comments is gathered under a dedicated title. We hope this will ease the reviewer's reading.

**General comment #1**

*#I have a few concerns about the idealized internal tide modeling. In particular, I am not sure of the utility of the chosen model, especially because its current implementation is not well-explained in the current manuscript.*

>>The model used in this study does not include any background currents and only resolves the tidal currents, pressure and free surface for the ITs generation and propagation. Some modifications have been made in the introduction and the model description sections to emphasize this. See the answers to the related comments below.

*#On Line120, the authors state: "Because the ocean circulation affects the ITs propagation, the complexity of its impacts on the ITs is beyond the scope of this study. Even though the stratification will be derived from the circulation, the stratification will be investigated as stationary in order to prevent further interaction with the circulation." These are very confusing sentences.*

>>Reformulation:

"In addition of the stratification, the currents also affect the ITs propagation and complexify the ITs signal. The investigation of such dynamical impacts over ITs is beyond the scope of this study. Here, the stratification is investigated without background current in order to only quantify the ITs signal response to the stratification alone. Such stratification is further named background stratification."

*#The second motivation difficulty is that the distinction between two regions is attributed to differences in factors controlling stratification. Namely, the authors suggest the significance of solar radiation compared to geostrophic currents controlling stratification. But then in all of the simulations, the stratification is horizontally uniform. So, how are the effects of currents on stratification actually retained in these results?*

>>The effect of currents on stratification is retained because the realistic stratification profiles are the sum of every processes taking place and the currents are one of them. The current-driven stratifications are used to investigate the response of ITs without any other complexification in order to properly quantify it. The simulations enable to prove that the waters inside the core of the NBC rings (deeper pycnocline) implies a longer wavelength of the ITs than outside the NBC rings (shallower pycnocline). We choose to not deepen the interpretation because the ITs also dynamically interact with the currents and the simulations are not designed for that as there is no background currents.

*#My final general comment concerns the use of the tidal model in this study. I think there needs to be much more care in this section. The choice of bathymetry (Equation 4) seems very generic, while the two*

*basins shown in figure 2 are very different. How relevant is this choice of idealized bathymetry to either region? Perhaps comparing a transect of bathymetry from figure 2 to equation 4 would be useful.*

>>This generic bathymetry enables to simulate really clean ITs and to only focus on the stratification impacts only. A new figure about the bathymetry comparision is added in the appendix. Following sentence is added in the model description (l.456):

"This bathymetry is similar to an averaged continental slope, a comparison to realistic bathymerty of the two areas of interest is shown on Figure C1."

*#If the overall point that the authors want to make concerns the variability in IT wavelength at multiple vertical modes, a linear eigenvalue solution to the stratification profiles would give that result without needing this idealized model.*

>>Such approach could provide the wavelength but not the amplitude of the ITs. As shown on the figure 7 and 9, the amplitudes of the modes are also highly affected by the stratification.

*#I do not recommend the authors use the Nugroho, 2017, reference as a primary citation for 3D T-UGO model configuration or for the modal decomposition as this work has not been through peer-review. Instead, if these modelling results are to be used, many more details can be provided: What are the equations solved, boundary conditions, and solution procedure?*

>>Actually, there is no other references that describe 3D T-UGOm. For short, it solves for the quasi-linearized, frequency domain 3D equations, formulated in the 3D, vertically lagrangian, equivalent of the well-known 2D wave equation. Level displacements and density anomalies (due to advection), governing pressure anomalies, are the primary unknowns. The discretized equations form a linear, complex-valued system which solution is obtained through a single inversion. Once solved, horizontal velocities are obtained by the use of the horizontal momentum equations. However, non-linearities (such as bottom friction/vertical momentum diffusion) are solved by iterating the tidal solver with non-linear terms set from the previous inversion. Boundary conditions are formulated to prescribe the barotropic tidal component and a buffer zone is implemented to avoid internal tide reflection at open boundaries. A complete article is in preparation, based on Nugroho (2017) and the additionnal developpements that have been made since (Lyard et al.). The mention to such article is added to l.429:

"Initially developed to resolve the two dimensional tidal equations (Piton et al., 2020 ; Lyard et al. 2021), this model has been extended to resolve the three dimensional tidal equations in the frequency domain (Nugroho, 2017; Lyard et al., in prep). "

**General comment #2**

*#Although overall the analysis and figures are engaging, I found much of the language used to be awkward and at times misleading. I strongly recommend additional editing of the language before resubmission.*

>>The article will be read by an english-native in order to improve the language.

**Specific Comment #1**

*#My first comments concern the motivation of the work. If one of the motivations are to quantify internal tide non-stationarity, this is not done. Stratification profiles are examined, and non-stationarity is inferred, but don't the realistic models, HRET V8.1 and NEMO (line 605) include non-stationary tides? Why not use the 2D FFT method to actually address the non-stationarity effects?*

>>HRET V8.1 only represents the stationary part of ITs and the NEMO simulations does represent both stationary and non-stationary parts. But the non-stationary part is extremely complex and important. The methods to extract it are very sensitive to the frequency windows used. The quantification of the ITs non-stationarity requires a dedicated study that is in preparation using NEMO outputs (Tchilibou et al.). Here, the motivation is not to quantify the ITs non-stationarity but to improve the comprehension of the ITs variability, to help the community to precise their interpretations of ITs non-stationarity. For this purpose, we focus on the impacts of only one parameter, the stratification, and quantify its impacts on both ITs amplitudes and wavelengths. Reformulation of the goal in the introduction (l. 70):

"The present study aims at contributing to the understanding of the ITs' non-stationarity through the investigation of the ITs variability".

*#In Section 3.3, I believe Figure 10 is not what you want to show?*

>>Indeed, the figure 10 was a duplicate of figure 9, this has been corrected, sorry for the embarrassment. The figure and the text are updated to consider all the 5 clusters (not only AS-108m) for the comparison (l.670):

"The wavelengths in HRET and NEMO are coherent with both modes 1 and 2 wavelengths calculated from the clusters, but models' wavelengths are slightly longer than the averaged cluster wavelengths. As explained in the introduction..."

**Specific Comment #2**

*#My second comment is on the profile clustering methodology. I have a few suggestions that can clarify: Line 205 - 235: The description here should be improved. For example, why are there only 2 coordinates in the PCA?*

>>Addition of few sentences to clarify this point:

"As the profiles are only described by the density versus depth, only two principal components are used. Thus, the shape of each profile is evaluated according to two orthogonal axes. The PCA manifold is the plan defined by these two new axes and where each profile is characterized by a point. The both axes explain a different part of the standard deviation of the profiles. For example, if the profiles are mainly controlled by the pycnocline depth but also by the surface density, the first axis (PC1) will be controlled by the different depths of the pycnocline whereas the second axis (PC2) will be controlled by the surface depths."

*#Section 2.2.2. discusses some optional parameters in the clustering, but what is the effect? In particular, I am not sure how the authors determined "the best results (line 235)" and why the Ward method would have a stability criteria? Perhaps this sensitivity analysis can be moved to an appendix*

*that includes a few of the examples written in words here, but instead portrayed graphically so that the reader can follow?*

>>Thank you for this advice, most of the section 2.2.2 (now section 3.2.2) have been moved to the appendix A where 3 additional figures illustrate the discussions. The section has been rewritten as follow:

"Three methods of clustering are compared: Ward, Average and Spectral. Those methods have been selected because they can better classify similar PCA manifold that we have. For each method, the sensitivity of two parameters needs to be investigated: the number of final clusters and the number of neighbors used in the calculation of the distance between profiles. The number of neighbors is important to properly manage the profiles that are isolated outside the PCA manifold. If the number of neighbors is weaker than the number of outsider profiles, then they all would be grouped in a dedicated cluster. Otherwise, they would be included in the cluster of the nearest profiles. This latter case can lead to groups of profiles that do not have the same shape inside the same cluster. The number of neighbors also affects the profiles located at the boundary between two clusters: depending on the number of neighbors, they would be included in one cluster or another.

A wide range of sensitivity tests have been made to choose the best method and the best parameters. The number of neighbors is tested from 4 to 16 and the number of clusters is tested from 2 to 10. These results can be found in the appendix A. The Ward method is used in the rest of the study because it offers a wider range of stratification cases and it is less sensitive to the number of neighbors. The classifications using the 16 nearest neighbors are distributed more equally between the clusters so this parameter is chosen. The number of clusters is set more arbitrarily. For the western equatorial Atlantic, the variability of the density profiles is controlled by the pycnocline depth with almost no modification of the N profile. Thus, few clusters are needed to characterize such variability. For the Bay of Biscay, the variability of the density profiles is way more complex and N profiles are very different even for 10 clusters. But a high number of clusters leads to have some clusters with few profiles. Thus, for both areas, a classification of 6 clusters is a good compromise that enables us to detail the evolution of the density profiles while keeping well represented clusters (more than 100 profiles)."

**Specific Comment #3**

*#Line 620: "The bathymetry of the T-UGOm simulations is set capped to 4000m whereas the real bathymetry in the area can extend down to 4500 m in the generation zone and down to 5000m further north" How do the authors know that internal tides are generated at these depths?*

>>The term "generation zone" is misleading. The sentence is corrected as follows:

"The bathymetry of the T-UGOm simulations is set capped to 4000 m whereas the real bathymetry in the area can extend down to 4500 m close to the continental slope and down to 5000 m further north."

**Technical corrections**

*#Please rewrite the sentences on line 275. They are very confusing and grammatically incorrect: "The temporal variability of the clusters (Fig. 3b,e) shows that every cluster happen all the year. There is a seasonality very noisy due to the complexity of the circulation, its spatial distribution and its seasonality*

*(explained below). The cluster classification enable to focus on a simple parameter (the pycnocline depth) rather than being blurred by the noise of a classical seasonal average classification. "*

>>Reformulation :

"The clusters are not strictly defined during a specific period of the year but rather during all along the year (Fig. 3b,e). In addition, the spatial distribution of the clusters is not homogeneous within the area highlighting spatially-bound ocean processes responsible for some specific stratification. As the pycnocline depth is highly controlled by the circulation, the complex spatio-temporal variability of the clusters refers to the complex spatio-temporal variability of the circulation in this region. The clusters classification enables to focus on a simple parameter (the pycnocline depth) that would be smoothed with a classical seasonal average classification."

*#Line 375: Missing a period "... uniform horizontally There, the cluster..."*

>> Corrected.

*#Line 375: I don't understand the use of the word "concise" in this sentence.*

>>Reformulation :

"There, the 6 clusters classification gathers the same amount of information about the seasonality than the 12 groups of the monthly classification. Thus, the cluster classification is a more condensed approach."

*#Line 380: I don't understand what the relevance of these statements. Can you reframe?*

>>Reformulation :

"As the cluster analysis does not preferably consider time dependent or space dependent classification, this method is very effective to investigate circulation-driven stratification variability, such as in the tropics."

*#Line 385: What is the relevance of observing long-term variability here?*

>> Addition of the following sentences:

"In a classical seasonal or monthly averaged classification, such long-term variability would have smoothed the stratification profiles. Here the clusters mean density profiles are based on similar instantaneous profiles, insuring more realistic profiles."

*#Line 390: "Grid"*

>> Corrected.

*#Line 400: "This enables us to compare the simulations with realistic cases." What cases are you referring to?*

>>Reformulation :

" This enables us to compare the simulations with ITs measurements and realistic simulations."

*Line 630: I am not sure that what the authors propose here would work. Wouldn't the addition of a mesoscale create non-uniform horizontal stratification? How would a cluster analysis help in that situation?*

>>Yes but the clustering methods can detect non-uniform horizontal stratification (like in the western equatorial Atlantic). Then, this spatially-bound stratification could be used to create spatially-bound ITs wavelengths maps over abyssal plains and ITs amplitudes maps over generation areas.

*Line 670: "The definition of a good parameter controlling the ITs amplitude and wavelengths need to be pursued in mid-latitude to unify the processing of the different regions of the global ocean." I do not understand this sentence. Can you rewrite?*

>>Reformulation :

"The efforts to find a formulation to link the ITs amplitude and wavelengths to the stratification need to be pursued for the mid-latitudes. is to obtain a parametrization that could unify the different regions of the global ocean.

*Figure 8 caption: I don't understand this, please reword: "...the colored patches represent the part of each mode in the sum: the modes on top of the sum line refer to destructive interaction between the modes."*

>>Reformulation on both Figure 6 and 8 :

"...the colored patches represent the modal contribution to the complex sum: if the patch of mode *n* is located on top of the sum line, then mode *n* works against mode *n-1*."

---

## Author Response (AR2)

Dear Pr. Fer,

We do thank you for your careful reading of our manuscript and relevant comments. A strong revision of the manuscript have been made to shorten it, clarify the objectives and highlight the main results. Additional results about the ITs energy flux have been produced following the advises of RC#3 and yours. Additional validation of the simulation have been made following the advises of RC#2. The results from Tchilibou *et al.* (to be submitted) about the seasonal study of ITs in the NEMO simulation are provided in advance thanks to the authors and will be submitted to Ocean Science journal by September 2021. The appendix about the sensitivity of the clustering methods have been moved to supplementary material as the methods are not new and the description quite long.

Below your comments are listed (after #) and followed by our answers (after >>). The modifications are provided when they are not too long (between "..."). Changes made in the manuscript are highlighted in red and blue in the revised version of the manuscript. We took the liberty to reorganize the document by gathering together the comments tackling the same issue, in order to make our answers clearer. We hope this will ease the your reading.
* * *
**I strongly support reviewer #3's suggestion on a discussion of baroclinic energy fluxes.**
>> A subplot is added on the figure 6 and 8 (previously fig.7 and 9) to show the energy flux of each simulation for the five first modes. To meet more realistic energy fluxes, the barotropic tidal current have been reduced to 5 cm/s. This modification does not change the conclusions highlighted in the study and only affect the amplitude of the values. The discussion about the comparison between the simulation amplitudes and the ITs altimetry atlases have been updated consequently.
Energy flux value have also been used to validate the T-UGOm simulations (WEA) comparing them with NEMO simulation (data from Tchilibou et al., to be submitted).

**There are repetitions and basic textbook information which can be reduced substantially. I made some suggestions and give examples below, but these are not exhaustive. Please attempt similar cuts where needed. Line numbers refer to manuscript-version3.pdf. I stop my comments at Section 4.2 and expect you to make similar improvements and clarifications throughout.**
**Abstract is too long, and in parts not clear (e.g. last sentence can be rewritten). Some other suggestions are below.**
>> Many parts have been removed and/or reformulated to improve the reading.

**Li3, not clear to the reader why the repetitivity of the orbit implies what you claim.**
**Li1-3 could be shortened, e.g. «The forthcoming SWOT altimetric missions aim to resolve the mesoscale with an unprecedented spatial resolution and swath, but high frequency processes, such as internal tides (ITs) are undersampled in time and aliased onto lower frequencies.»**
>> The SWOT mention have been shorten as suggested.

**Li 9-12, can be shortened: "Here we present a method to quantify the impacts of background stratification using a clustering method for classification of a broad range of stratification and idealized modelling of the ITs in frequency domain." (avoid T-UGO which needs introduction, and there's no room in the abstract)**
>> Done

**Li 15-16, for example: "…increases the total ITs' amplitude, transfers energy from mode 2 to mode1, and increases the wavelengths of both modes. In the …"**
>>Rephrased: "For the western equatorial Atlantic, a single pycnocline is observed and only the two first vertical modes of ITs have significant amplitudes. The depth of this pycnocline linearly impacts on the amplitudes and wavelengths these two modes. An increase of the pycnocline depth increases the total ITs' amplitude but also transfers the energy from mode 2 to mode 1."

**Li 17-18: "…of modes 2 and 3 and the surface elevation of ITs. On the other hand, the wavelengths…."**
>>The results about the wavelength have been removed as they are not the main point of the article (in agreements to the comments of reviewer #2)

**Avoid use of «significant» when not related to statistical analysis (e.g., li6, li18, where «substantial» could be an alternative)**
**Throughout: care with use of "e.g." (see lines 37-38, all e.i.'s must be e.g.)**
>>Done

**Li59- coarse temporal resolution**
>> Done

**Li 53: is T/P introduced?**
>>Corrected

**Li 76, reword,"after a brief introduction to Its and stratification"…(but I recommend you heavily reduce this brief introduction).**
**Li 122-130: the entire set of two paragraphs can be cut out.**
>>The section « Background knowledge » have been removed as well as the figure 1. The introduction have also been reworked and shorted to clarify the objectives and the approach.

**Eq.1 : This equation is not correct and is definitely not what is implemented in TEOS-10. In li173 you say you are using TEOS-10 for calculation of potential density profile. Are you also using it to calculate Nsquared? If so, give the TEOS-10 equation (and also with potential density approximation) for Nsquared (see their manual). Your rho0 appears to be pot density referenced to surface pressure. It is very important to note that (when working with water depths of several km) the potential density used in the gradient must be referenced to local pressure when calculating Nsquared.**
**Fig 2 caption: What do you mean "in situ" density profile when you prefer potential density anomaly? Must also mention the reference pressure. How is N2 calculated? Using the mean density profile?**
**Li170-175. Readers of Ocean Science know what "potential density" is. Cut out and simplify this to: "Potential density and N are calculated using the TEOS-10 ……"**
>>The figure 1 (previously fig 2) now shows the buoyancy frequency as calculated in TEOS-10, the equation have been removed and the text have been reworked.

**Li 140: plural extramum is extrama**
>>Also corrected everywhere else.

**Fig 2. There's a sharp change in N2 at 1500 m depth for both sites. I am sure this is not natural and is an artefact of your smoothing filters.**
**Li 245-250: all this filtering appears confusing, not well justified (especially see the profile change at 1500 m). A simple sorting of the median profile, and them vertical smoothing over 25 m or so would have been much simpler (or perhaps a spline fit to selected data points of the profile).**
>> Yes, the smoothing method have been modified, thank you for pointing that out.

**Li 193-[statically] stable density**
>>Done

**Li 196-202: need substantial cut and simplification. Example: "All profiles with more than 5 measurements over upper 100 m, where the stratification is most variable, are linearly interpolated to a uniform 1 m vertical resolution, and processed using the principal….."**
>> The section about numerical processing of the profiles have been reworked and shortened.

**Li 239: …profiles must be stable and defined…**
>>Done

**Li 243: of the deepest 4 measurements**
>>Done

**Li 263: This reads like you "reserved" the 6th group to suspicious profiles, but this is not the case for BoB. Please reword.**

>>Rephrased: "As these profiles are less numerous, they will form the last cluster
of the classification (WEA-6). Form now on, the cluster WEA-6 is neglected and only the realistic profiles
contained in the other clusters are considered."

**Li 283: plum[e]**
>>Also corrected everywhere else.

**Fig 3a, PC1 vs PC2. There are outliers, all WEA-6 and a few WEA-1. Is it an idea to use the scatter plot to**
exclude such profiles, without claiming one (the 6th) group separates the suspicious profiles?
>>The scatter plot helps to understand how the clustering method gathers neighbour profiles/group of
profiles. The outliers of WEA-6 are easily identified and are numerous enough to form a cluster. At the
opposite, the outliners of WEA-1 are not numerous enough to form a cluster on their own.

**Fig 3 caption. Isobath contours are gray (not black)**
>>Done

**Li 278-309: this part can be reduced substantially. Also some comments on the stratification made earlier in**
**li 177-185 can be omitted (from li 1777-185), since they are repeated here.**
>>Previous mention of NBC removed and section shortened.

**Li 317: highlight that**
>>Done

**Fig 4. a) again there're some outliers (in BB-5). It could be an idea to exclude from analysis? Caption:**
delete the entire text and use: "Same as Figure 3 but for the Bay of Biscay shelf."
>>The outliers in the Bay of Biscay profiles are way closer to the main groups than the one in the western
equatorial Atlantic profiles. In addition, there is only 2 profiles concerned so, there are not numerous enough
to form a cluster. Exclude them won't change the classification.

**3.4. Discussions- some of the descriptions from previous sections (on stratification variability and its**
causes) are repeated here. It can be better organized here (and please avoid repetitions)
>>Some sentences have been removed and reworked.

**Li 350-352: reduce to "V4.3 seasonal profiles, likely due to the spatial and vertical smoothing used in the**
climatology."
>>Done

**Li 411-423: delete the sentence "As explained earlier…"**
>>Done

**Li 436-443: The entire paragraph can be reduced to " In the vertical, 80 sigma-layers (which follow the**
bathymetry) are used, with increased resolution in the upper ocean, set by a cosine function between 0 and
pi/2" (please correct and improve my interpretation)
>>Rephrased: "On the vertical, 80 sigma-layers (which follows the bathymetry) are distributed using a
cosine function between 0 and pi/2. This vertical distribution enable to better represent surface pycnocline
and the associated ITs than a uniform distribution."

Dear Reviewer #1

We do thank you for your careful reading of our manuscript and relevant comments. A strong revision of the manuscript have been made to shorten it, clarify the objectives and highlight the main results. Additional results about the ITs energy flux have been produced following the advises of RC#3, supported by the editor. Additional validation of the simulation have been made following the advises of RC#2. The results from Tchilibou *et al.* (to be submitted) about the seasonal study of ITs in the NEMO simulation are provided in advance thanks to the authors and will be submitted to Ocean Science journal by September 2021. The appendix about the sensitivity of the clustering methods have been moved to supplementary material as the methods are not new and the description quite long.

Below your comments are listed (after #) and followed by our answers (after >>). The modifications are provided when they are not too long (between "..."). Changes made in the manuscript are highlighted in red and blue in the revised version of the manuscript. We took the liberty to reorganize the document by gathering together the comments tackling the same issue, in order to make our answers clearer. Some of your comments are no longer relevant because the sentences have been removed, all these comments are gathered at the end of the document. We hope this will ease your reading.
* * *
**l. 56: 'since the tidal currents are not in geostrophic balance.'**
**l. 130: Sentence 'Such stratification…' repeated twice (second time with italics)**
>> Corrections included in the reworked introduction

**l. 77: 'Section 2 addresses the usage…' what do you mean by that?**
>>Rephrased:"Section 2 details the clustering method and compares its results to the classical three-month mean. Section 3 details the modeling of the ITs based on the typical profiles obtained from the clusters."

**l. 138: '… is to consider either seasonal means (the mean of three months…), or…'**
**l. 176: replace 'both' by 'the two'**
**l. 235: '… that enables us to discuss…'**
>> Corrections included in the reworked methodology of the clustering.

**l. 170: 'The potential density is calculated for all data…' Then, 'The potential density is …, which is equivalent to …'**
**l. 172 and 173, replace 'convention' by 'definition' or something else more appropriate.**
>>Rephased: "The potential density and bouyancy frequencies are calculated from TEOS-10 definitions…"

**l. 204: 'only two principal components are used…' : what does it mean: there are many depths…**
then line 209: '… by the surface depths'?
Maybe further explanations are required on the PCA method used, as this is not the most standard one, when it would be an analysis of T(P) for all the individual profiles.
>> The PCA is only a method that enable to calculate the distance matrix, needed to perform the cluster analysis. Further explanation are given in the supplementary material about the clustering methods sensitivity: "The clustering methods can classified the different profiles in several clusters using a matrix of the distances from every profiles to the other ones. Performing a principal component analysis (PCA) with two principal components on the profiles enable us to transform a system of (NxD) (with N the number of profiles and D the number of depths) to a system of (Nx2). In such system, named the PCA manifold, the distance matrix is easily calculable (Fig. A1)."

**l. 214: 'directly affected by the value of the density r0'… This is a minor effect, not a 0-order one, and I would not mention it. There is virtually no information lost by applying a normalization, contrary to what is stated here.**
>> The normalization by the mean of the profile affect the profiles with an offset like the one in WEA-6 and inhibit the clustering method to consider them as outliers. These sentences have been removed in the shortened version of the methods.

**l. 257: '… and were measured during the same period…'**
>> Done

**l. 260: '… for cycles 150-152 and for cycles 166-180 (except cycle 176)**
Note that these should not have passed the standard quality control, based on min-max applied on the CORA data set (but you use a fairly old version; note that even for recent versions, the individual 'validated/qualified' profiles are available, and not just the gridded product, contrary to what is stated at the beginning)
>> Done. Thank you for the details, a more recent version of CORA will be used for other analysis.

**l. 272: '… defined during a specific season, but rather during the whole year'**
>> Done

**l. 304: '… the difference in pycnocline depth…'**
>> Done

**l. 326: 'to build a surface stratification without…'**
>> Done

**3. 4 'Discussion'**
>> Done

**l. 335: 'The stratification in the …'**
>> Done

**l. 337: '… are limited-extent processes, whereas…'**
>> Rephrased: "The stratification variability due to the circulation is spatially bound, whereas the one due to radiative forcing affects the area homogeneously."

**l. 343-344-346-347: 'is compared with…'**
>> Also corrected everywhere else.

**l. 348: '… are averaged in the same areas…'**
>>Done

**l. 354: '… show…'**
>>Done

**l. 360: '…, as well as the cluster classification do.'**
>>Done

**l. 361: '… is not effective.'**
>>Done

**l. 365: '… profiles have.'**
>> Corrected: "profile has..."

**l. 445: maybe '… in order to fit the bottom velocity boundary condition'.**
>> Rephrased: "Using a frequency domain calculation, there is no spin-up of the simulation that could lead to a stable value of the bottom friction."

**Figure 6 and Fig 8 captions, line 3: '… show the extrema of the modes.'**
>>Done

**l. 462: 'We will first discuss the vertical structure of the modes for…'**
>>Done

**l. 555: '…, while mode 2 is controlled…'**
>>Done

**l. 559: '…, but have the right magnitude. The difference could originate from differences in…'**
>>Done

**l. 585: '… seems to …'**
>>Done

**4.3 'Discussion'**
>> Renamed: "Validation and discussion"

**l. 604: 'The Its simulations…'**
>>Done

**l. 608: '… have been equally set in the two configurations.'**
>>Done

**l. 615: I would not call the western equatorial Atlantic' 'the amazonian one'**
>> Yes, it was a missing correction from the previous reviews.

**l. 638: 'This almost perfectly fits the wavelength of HRET and NEMO.'**
>>Done

**l. 654: '… is an improvement compared with the mean…'**
>>Done

**l. 19: sentence should not start by Whereas…**
**l. 26: '… enhance vertical mixing in the ocean.'**
**l. 31: 'Their method…'**
**l. 36: replace 'as well as …' by 'as well as ocean currents'**
following lines: 'for a realistic approach' 'for an idealized approach'
**l. 64: 'The SWOT measurements will face two issues: …'**
**l. 142: 'To maintain the realism of the typical profiles with only a few of them…'**
to be replaced by 'To extract a limited set of profiles that are representative of the whole set of profiles, we will use clustering methods.'
**l. 169: '… is also built…'**
**l. 193: senetence 'The measurements…' unclear. What is meant by 'unstable water masses'?**
**l. 200: remove 'Now that the profiles are properly selected and interpolated,'**
**l. 206 'Both axes…'**
**l. 353: '… the fall profiles show… than the spring ones'**
**l. 283: 'In addition to the water masses…'**
**l. 403: 'The concerns affect the…' I am not sure that I understand the sentence**
**l. 410: 'As earlier explained, …'**
**l. 437: '… also needs to be investigated.'**
>> Removed sentences

Dear Reviewer #2

We do thank you for your few but relevant comments. A strong revision of the manuscript has been made to shorten it, clarify the objectives and highlight the main results. Additional results about the ITs energy flux have been produced following the advises of RC#3, supported by the editor. Additional validation of the simulation has been made following your advises. The results from Tchilibou *et al.* (to be submitted) about the seasonal study of ITs in the NEMO simulation are provided in advance thanks to the authors and will be submitted to Ocean Science journal by September 2021. The appendix about the sensitivity of the clustering methods have been moved to supplementary material as the methods are not new and the description quite long. We hope this new version will meet your expectations.

Below your comments are listed (after #) and followed by our answers (after >>). The modifications are provided when they are not too long (between "..."). Changes made in the manuscript are highlighted in red and blue in the revised version of the manuscript. We hope this will ease the your reading.
* * *
**1. In response to reviews, the authors state: "The quantification of the ITs non-stationarity requires a dedicated study that is in preparation..." and yet in this manuscript "The present study aims at contributing to the understanding of the ITs' non-stationarity through the investigation of the ITs variability." These two statements are at odds. I suggest the authors state that this manuscript concerns seasonal variability in IT and do not discuss non-stationarity.**
>> They are not. The non-stationarity of the ITs in the altimetry is caused by temporal stratification variability as well as the interaction between the internal waves and the oceanic circulation. This study only focuses on the part of the ITs non-stationarity due to the temporal variability of the stratification. The sentences have been reworked.

**2. "The wavelengths in HRET and NEMO are coherent with both modes 1 and 2 wavelengths calculated from the clusters." In the case that you are concerned about wavelengths, there is no reason to run TUGO. Wavelengths are simply arrived at from solutions to a linear eigenvalue problem. Without seasonally-varying amplitude comparisons in HRET and NEMO to the TUGO solutions, and demonstrating that the TUGO solutions are sensible, I do not see the point of running the TUGO simulations. Particularly with a model implementation that has not been peer-reviewed.**
>> This comparison is impossible to do with HRET or any other ITs atlases built from altimetry observation as there is no seasonal processing of the altimetry that have been developed so far. In NEMO simulation, the ITs properties have been investigated for two seasons by Tchilibou et al. (to be submitted). We used the elevation amplitude and the integrated energy flux of the ITs of these two seasons to compare them with corresponding T-UGOm simulations (Tab. 3).

Dear Reviewer #3

We do thank you for your careful reading of our manuscript and relevant comments within such short notice. A strong revision of the manuscript have been made to shorten it, clarify the objectives and highlight the main results. Additional results about the ITs energy flux have been produced following your advises, supported by the editor. Additional validation of the simulation have been made following the advises of RC#2. The results from Tchilibou *et al.* (to be submitted) about the seasonal study of ITs in the NEMO simulation are provided in advance thanks to the authors and will be submitted to Ocean Science journal by September 2021. The appendix about the sensitivity of the clustering methods have been moved to supplementary material as the methods are not new and the description quite long.

Below your comments are listed (after #) and followed by our answers (after >>). The modifications are provided when they are not too long (between "..."). Changes made in the manuscript are highlighted in red and blue in the revised version of the manuscript. We took the liberty to reorganize the document by gathering together the comments tackling the same issue, in order to make our answers clearer. Some of your comments are no longer reverent because the sentences have been removed, all these comments are gathered at the end of the document. We hope this will ease the your reading.
* * *
**What is the difference in (horizontal) IT energy flux between the different cases? I assume that is an easy think to compute from the model and it would be interesting to see a figure of that. After all, the ITs themselves are not what matters for the Earth system, it is the energy they redistribute that matters. Adding this would add value to the study and make the title more valid.**
>> A subplot is added on the figure 6 and 8 (previously fig.7 and 9) to show the energy flux of each simulation for the five first modes. To meet more realistic energy fluxes, the barotropic tidal current have been reduced to 5 cm/s. This modification does not change the conclusions highlighted in the study and only affects the amplitude of the values. The discussion about the comparison between the simulated amplitudes and the ITs altimetry atlases have been updated consequently.
Energy flux value have also been used to validate the T-UGOm simulation comparing them with NEMO simulation (data from Tchilibou et al., to be submitted).

**Typos need to be fixed.**
L39: e.i. should be e.g. (for example).
>>Done

**L130: freshwater is another controller of stratification, completely dominant in some locations and should be mentioned. This is especially key for western Atlantic region being discussed (cf. L364 where the amazon plume suddenly appears).**
>> Correction included in the reworked introduction.

**L508: "was expected weaker"; rewrite or clarify. Do you mean as expected?**
>> Competed: "whereas WEA-88 m was expected weaker to be sorted between WEA-70 m and WEA-108 m like for mode 1 and 2."

**Figure 7 and its discussion in the text: clarify that you mean horizontal wavelength.**
>> Done